# Testing and Development of Transfer Functions for Weighing Precipitation Gauges in WMO-SPICE

John Kochendorfer[1], Rodica Nitu[2,3], Mareile Wolff[4], Eva Mekis[2], Roy Rasmussen[5], Bruce Baker[1], Michael E. Earle[6], Audrey Reverdin[7], Kai Wong[2], Craig D. Smith[8], Daqing Yang[9], Yves-Alain Roulet[7], Tilden Meyers[1], Samuel Buisan[10], Ketil Isaksen[4], Ragnar Brækkan[4], Scott Landolt[5], and Al Jachcik[5]

[1]Atmospheric Turbulence and Diffusion Division, ARL, National Oceanic and Atmospheric Administration, Oak Ridge, TN, 37830, US
[2]Environment and Climate Change Canada, Toronto, ON, M3H 5T4, Canada
[3]World Meteorological Organization, Geneva, CH-1211, Switzerland
[4]Norwegian Meteorological Institute, Oslo, 0313, Norway
[5]National Center for Atmospheric Research, Boulder, CO, 80305, US
[6]Meteorological Service of Canada, Environment and Climate Change Canada, Dartmouth, NS, B2Y 4N6, Canada
[7]Meteoswiss, Payerne, CH-1530, Switzerland
[8]Environment and Climate Change Canada, Saskatoon, SK, S7N 3H5, Canada
[9]Environment and Climate Change Canada, Victoria, BC, V8P 5C2, Canada
[10]Delegación Territorial de AEMET (Spanish National Meteorological Agency) en Aragón, Zaragoza, 50007, Spain

*Correspondence to*: John Kochendorfer (john.kochendorfer@noaa.gov)

**Abstract**

Weighing precipitation gauges are used widely for the measurement of all forms of precipitation, and are typically more accurate than tipping-bucket precipitation gauges. This is especially true for the measurement of solid precipitation; however, weighing precipitation gauge measurements must still be adjusted for undercatch in snowy, windy conditions. In WMO-SPICE (World Meteorological Organization Solid Precipitation Intercomparison Experiment), different types of weighing precipitation gauges and shields were compared, and adjustments were determined for the undercatch of solid precipitation caused by wind. For the various combinations of gauges and shields, adjustments using both new and previously existing transfer functions were evaluated. For most of the gauge and shield combinations, previously derived transfer functions were found to perform as well as those more recently derived. This indicates that wind shield type (or lack thereof) is more important in determining the magnitude of wind-induced undercatch than the type of weighing precipitation gauge. It also demonstrates the potential for widespread use of the previously developed transfer functions. Another overarching result was that, in general, the more effective shields, which were associated with smaller unadjusted errors, also produced more accurate measurements after adjustment. This indicates that although transfer functions can effectively reduce measurement biases, effective wind shielding is still required for the most accurate measurement of solid precipitation.

## 1 Introduction

Precipitation measurements are frequently underestimated due to the interactions among wind, the precipitation gauge, and hydrometeors in the air around the gauge. Wind deflected over and around a precipitation gauge can alter the trajectory of hydrometeors falling toward the gauge inlet, diverting them away from the inlet and causing the gauge to underestimate the actual precipitation rate. The magnitude of this underestimation is affected by the wind speed and the phase, size, density and crystal habit of precipitation, and is largest for snowy, windy conditions. Many past observational (e.g. Rasmussen et al., 2012; Wolff et al., 2013; Ma et al., 2015; Wolff et al., 2015; Chen et al., 2015) and theoretical (Theriault et al., 2012; Colli et al., 2015; Colli et al., 2016; Nespor and Sevruk, 1999; Sevruk et al., 1991; Baghapour et al., 2017) studies support this finding, including the first World Meteorological Organization (WMO) Solid Precipitation Measurement Intercomparison performed in the 1990s (Goodison et al., 1998;Yang et al., 1998b;Yang et al., 1995). To address this issue, adjustments (or transfer functions) were developed to correct the undercatch for manual precipitation measurements. These adjustments were typically a function of wind speed and precipitation type, as manual precipitation measurements are generally accompanied by manual precipitation type observations (e.g. Goodison, 1978;Groisman et al., 1991;Yang et al., 1998a;Yang et al., 1999;Yang et al., 2005). More recently, transfer functions for automated precipitation measurements have been derived as a function of wind speed and air temperature (e.g. Kochendorfer et al., 2017b;Wolff et al., 2015). Transfer functions developed for automated measurements can be applied over shorter time periods, such as 30 – 60 min, whereas manual measurements are mainly adjusted per observation, with a typical observation period of either 12 or 24 hours.

Since the previous WMO Solid Precipitation Measurement Intercomparison (Goodison et al., 1998), many new automated sensors designed to measure solid precipitation have become available. In addition to not requiring a human observer, which allows them to be deployed in remote locations, automated measurements can be recorded at higher frequencies and can be used to monitor precipitation continuously throughout a storm. Instead of recording precipitation every 12 or 24 h, precipitation accumulation and rate can be accurately monitored in real time. The primary types of automated precipitation gauges available today are heated tipping-bucket gauges, weighing type gauges, and non-catchment gauges. Non-catchment type precipitation sensors, which typically monitor the intensity of precipitation using optical sensors, can also be used to record precipitation type. They can detect very low precipitation rates, but can also suffer from inaccuracies in measuring the intensity of solid precipitation over shorter time intervals (e.g. 30 minutes) due to variability in hydrometeor size, fall velocity, and density (Roulet et al., 2016). In addition, it is difficult to validate or calibrate a non-catchment type precipitation gauge. Heated tipping-bucket precipitation gauges are tipping-bucket gauges equipped with heaters to melt solid precipitation collected in the gauge inlet, allowing the tipping mechanism to measure liquid precipitation as discrete tips. Weighing precipitation gauges are also used to monitor all phases of precipitation, and they function by monitoring the total mass of precipitation collected below an inlet of known area. Weighing gauges should be serviced with antifreeze and oil to inhibit the freezing of the bucket contents, melt newly-collected precipitation, and inhibit the evaporation of antifreeze

and precipitation. Unlike heated tipping-bucket precipitation gauges, precipitation collected in a weighing gauge does not need to be heated and melted prior to measurement, and weighing gauges typically measure very light and very heavy precipitation more accurately than tipping bucket gauges. However, as the capacity, resolution, and measurement frequency increase, all weighing gauges are limited in the smallest amount of precipitation they are able to resolve, and their ability to discern precipitation from measurement noise. This limitation is especially important for snowfall, which is often associated with very low precipitation rates. Additional shortcomings of weighing gauges are their limited capacity and the need to replace and dispose of oil and antifreeze every time they are emptied.

To update the findings of the previous WMO Solid Precipitation Measurement Intercomparison, which focussed mainly on manual observations (Goodison et al., 1998), and to evaluate many of the automated precipitation sensors that are currently in use, the WMO Solid Precipitation Intercomparison Experiment (WMO-SPICE) was initiated in 2010. The goal of WMO-SPICE was to study and correct the effects of wind-induced errors on automated solid precipitation measurements, and also to evaluate new and existing precipitation and snow depth sensors in different configurations and climate regimes. Because snowfall and precipitation measurement methods vary from one region or country to another, and also because errors and biases in these measurements vary as a function of climate, meteorology, and local topography (e.g. Kochendorfer et al., 2017a, hereafter K2017a), one of the goals of WMO-SPICE was to include as many countries and testbeds in the intercomparison as possible. In addition to the standard national precipitation measurement systems evaluated at most testbeds, many of the WMO-SPICE testbeds included a common set of sensors; this included a reference weighing precipitation gauge shielded in a double-fence (the Double-Fence Automated Reference, or DFAR) and air temperature, wind speed, and optical precipitation sensors. The DFAR was modelled after the Double Fence Intercomparison Reference (DFIR), which was the manual reference configuration used in the previous WMO Solid Precipitation Measurement Intercomparison (Goodison et al., 1998), utilizing the same design for the two large, concentric wood shields that surround the gauge and inner shield. The DFAR is described in more detail in Ryu et al. (2016; 2012).

Past WMO-SPICE related work included the development of adjustment functions for tipping-bucket gauge measurements from the Spanish WMO-SPICE site (Buisán et al., 2017). Measurements from two WMO-SPICE test sites that pre-date the project were also used to describe and correct wind-induced undercatch for different types of wind shields (Kochendorfer et al., 2017b;Wolff et al., 2015). In addition, measurements from eight different WMO-SPICE sites were used to derive multi-site adjustments for single-Alter shielded and unshielded weighing gauges (K2017a). The results of K2017a indicate that despite some climate- or site-specific biases, multi-site adjustments (or transfer functions) can be used to effectively minimize the wind-induced undercatch of solid precipitation. Site host-provided measurements were used to derive the K2017a multi-site single-Alter shielded and unshielded precipitation transfer functions. WMO-SPICE also included several manufacturer-provided weighing precipitation gauges for evaluation, in addition to weighing gauges tested within other types of shields of more specific national and scientific interest. These previously unevaluated measurements have been

processed using standardized methods developed and implemented by WMO-SPICE, allowing for both the creation of new transfer functions and the evaluation of existing transfer functions derived from independent measurements.

The goal of this work was to test and recommend transfer functions for all of the previously unevaluated WMO-SPICE weighing precipitation gauges and shield configurations. In the present study, transfer functions were developed and tested using WMO-SPICE measurements from several types of weighing gauges and shields. These include gauge and shield types that have never been intercompared before, and for which no other adjustments are currently available. Previously derived adjustments were also applied to these measurements to test their applicability and efficacy for each of the gauge/shield types under evaluation, and also to test the hypothesis that for a given shield type, the same adjustments can be used to minimize wind-induced errors for different types of weighing precipitation gauges of a similar size and shape. Our hypothesis is that the type of shield (or the lack of a shield) is the primary determinant of undercatch. The alternative hypothesis is that every type of weighing precipitation gauge requires its own transfer function. The WMO-SPICE measurements were used to test both hypotheses by comparing transfer functions derived specifically for each gauge and shield with transfer functions derived previously using other gauges and measurements. WMO-SPICE included several manufacturer-provided unshielded and single-Alter shielded weighing gauges for which no transfer functions had previously been derived or tested. For these unshielded and single-Alter shielded gauges, the performance of newly derived transfer functions were compared to the performance of transfer functions from K2017a, which were derived using host-provided WMO-SPICE measurements from eight different testbeds. WMO-SPICE also included weighing gauges within the larger double-Alter, Belfort double-Alter, and Small Double Fence Intercomparison Reference (SDFIR) shields. For these larger shields, new transfer functions derived from these WMO-SPICE measurements were compared to transfer functions derived from measurements that predate WMO-SPICE (Kochendorfer et al., 2017b, hereafter K2017b). Based on all these evaluations, specific transfer functions are recommended for all of the weighing gauges and wind shields included in WMO-SPICE.

## 2 Methods

### 2.1 Intercomparison overview

Measurements from all WMO-SPICE sites for which weighing precipitation gauge measurements were available for evaluation are included in this study. All of the sites included in this evaluation used a DFAR as the reference configuration. The reference precipitation measurements recorded at these WMO-SPICE sites were compared to simultaneous measurements from the gauges under evaluation. Catch efficiencies (*CE*), defined as the ratio of accumulated precipitation reported by a gauge under test to that reported by the reference configuration, were calculated. Using the computed catch efficiencies and concurrent measurements of air temperature and wind speed, transfer functions were created for the weighing gauges under evaluation. The same set of measurements was also used to evaluate independently derived transfer

functions from K2017a and K2017b. Based on the results of these evaluations, transfer functions are recommended for all of the weighing gauges included in the study.

## 2.2 Precipitation measurements

Weighing gauges and shield configurations tested at six of the WMO-SPICE testbeds (Fig. 1) are included in this study. In addition, Figure 1 includes the percentage of the total annual precipitation that is solid, as estimated using either 30 yr. (1981 – 2010) climate normal or the best available proxy data for these locations. These values, which ranged from 25 – 45%, demonstrate the contribution of solid precipitation to the annual water balance. These sites are described in more detail in K2017a. The gauges were all evaluated during 2013-2015, with measurements during the two winter seasons (October 1 –

April 30) from this period considered in the present analysis. Measurements were recorded at either 1 min or 6 s intervals, and transferred to a central database at the National Center for Atmospheric Research (NCAR) in Boulder, CO. The Double Fence Automated Reference (DFAR), which was defined as the automated reference for WMO-SPICE, consisted of either an OTT Pluvio[2] or a Geonor T-200B3 within a DFIR shield, and was used as the reference for all of the measurements evaluated here (Nitu, 2012). The DFIR shield as described by Goodison et al. (1998) comprises two concentric, octagonal

fences constructed out of 1.5 m long wooden lath, with the outer shield having a diameter of 12 m, and the inner shield having a diameter of 4 m. For the DFAR, at the centre of the inner shield the weighing gauge is installed in a single-Alter shield. Typically, the top of the single-Alter shield and the inlet of the weighing gauge within the DFAR are at a height of 3 m, but at Weissfluhjoch and Haukeliseter they were installed higher than this (at 3.5 and 4.05 m, respectively) to prevent drifting snow from burying the shield and gauge.

The weighing gauge models included in this study are detailed in Table 1, and include the Sutron TPG, Meteoservis MRW500, MPS systém TRwS 405, Geonor T-200-MD3W (1500 mm capacity), Geonor T-200B3 (600 mm capacity), and OTT Pluvio[2]. The shield configurations of these gauges are provided in Table 2. The TRwS 405 (MPS systém, TRwS 405, Slovakia) has a heated 400 cm[2] orifice, a 750 mm capacity, and uses a strain gauge type load cell to measure the mass of

precipitation accumulated within its bucket (Table 1). It was provided by the manufacturer without a wind shield and tested at both the Haukeliseter and Marshall testbeds (Table 2, Fig. 2a). The MRW500 (Meteoservis, MRW500, Czech Republic) has a heated 500 cm[2] orifice and a 900 or 1800 mm capacity (with or without antifreeze), also employs strain gauges for weight measurement (Table 1), and was tested at the Marshall and Bratt's Lake testbeds. Both an unshielded (Fig. 2b) and a shielded gauge (Fig. 2c) were installed at each site (Table 2), with the small, manufacturer-provided shield constructed out of

fixed metal slats and attached to the same base as the gauge. An unshielded and single-Alter shielded (Fig. 2d) Total Precipitation Gauge (TPG) were provided by Sutron (Sutron, TPG, USA) and tested at the Marshall testbed (Table 2). The Sutron TPG uses a load cell to quantify the amount of accumulated precipitation, has a 914 mm capacity, an 8" diameter inlet (20.32 cm diameter, 324.3 cm[2] area), and was provided with a heater (Table 1). The T-200-MD3W (1500 mm Geonor)

from Geonor (Geonor, Norway) was tested at Marshall, Bratt's Lake, Weissfluhjoch, and Caribou Creek (Table 2), but only the gauge at Weissfluhjoch was provided with a heater (Table 1). The 1500 mm Geonor is based on the same design as the 600 mm and 1000 mm Geonor T-200B3 3-wire, vibrating-wire gauges, but it has a taller cover, taller bucket with increased capacity, and different vibrating wire transducers.

The double-Alter shield (Fig. 2d), which consists of a single-Alter shield surrounded by a second, larger (2.4 m in diameter) row of 40 cm long slats, was tested at CARE (Centre for Atmospheric Research Experiments) with an OTT Pluvio$^2$ (OTT Hydromet, Pluvio$^2$, Germany; Tables 1 and 2) and at Marshall with a Geonor T-200B3 (3-wire, 600 mm capacity, T200B, Geonor Inc., Norway; Tables 1 and 2). Both gauges included inlet heating, with the Pluvio$^2$ at CARE using the
manufacturer-provided heater and the Geonor at Marshall using the US Climate Reference Network heater (described in NOAA Technical Note NCDC No. USCRN-04-01). Based on the additional shielding provided by the double-Alter shield and also past studies (K2017b), the double-Alter shield is expected to perform better than the single-Alter shield (reduced undercatch), and to accumulate greater than 50% of the DFAR accumulation even in snowy and windy conditions. The Belfort double-Alter shield (Fig. 2f), which has the same sized footprint as the standard double-Alter shield, but with a lower
porosity (30% vs. the standard double-Alter porosity of 50%) and longer slats (46 cm long for the inner shield and 61 cm long for the outer shield) that do not taper like the double-Alter slats, was also tested at both CARE and Marshall (Table 2). The standard double-Alter slats also rotate freely, while the Belfort double-Alter slats employ springs to limit their travel within 30° of the vertical. Like the standard double-Alter, the Belfort double-Alter was tested with a Pluvio$^2$ at CARE and a 600 mm Geonor T-200B3 at Marshall (Table 2). Prior to this study, this shield had only been evaluated at the Marshall
testbed (K2017b). Based on that study, gauges within this shield are expected to accumulate almost as much precipitation as the DFAR, with typical catch efficiencies greater than 0.8 in snowy and windy conditions.

The SDFIR shield (Fig. 2g), which is 2/3 the size of standard DFIR shield and was designed to be more easily constructed out of commonly sized North American lumber, was tested only at the Marshall site (Table 2). Like the standard DFIR
shield, the SDFIR comprises three concentric shields. The wooden laths on the two-outermost concentric shields were 1.2 meters long, and the diameters of the two outer shields were 8.0 meters and 2.6 meters. The height of the inner wooden shield was 10 cm lower than the outer shield. A standard single-Alter shield, mounted at the same height as the gauge inlet and 10 cm lower than the inner wooden shield, was mounted around the gauge. Based on its design and a past study (K2017b), we expect this shield to be almost indistinguishable from the DFAR, except in high winds and snowy conditions,
where it will typically accumulate about 90% as much precipitation as the DFAR.

### 2.3 Wind speed and direction

Wind speed measurements were used to create 30-minute-average wind speeds for each site. Because transfer functions developed from both the gauge height and the 10 m height wind speeds were desired, and not every site included wind

measurements at both heights, the available 30-min measurements and the logarithmic wind profile were used to determine either the gauge height or 10 m height wind speeds, when necessary. These methods are described in detail in K2017a. For the TRwS 405 precipitation gauge at Haukeliseter, an additional screening for wind direction was performed based on the gauge's position relative to the DFAR; precipitation measurements with wind directions between 115° and 140° were excluded from the analysis due to wind shielding by the DFAR.

## 2.4 Quality control and selection of 30-min periods of precipitation

Precipitation measurements were recorded at either 6-s or 1-min intervals (Table 1). These 6-s and 1-min data were subject to the following quality controls: a range filter, to remove values that exceeded the capacity of the gauge; a 'jump' filter, to remove sudden changes in accumulation exceeding a specified threshold; and a Gaussian filter, to remove high-frequency noise. For transfer function development, the resultant 1-min (all 6-s data were aggregated to 1-min), quality-controlled measurements were then used to create 30-min datasets that included only periods of precipitation. To exclude noise and light precipitation, a minimum threshold of 0.25 mm of precipitation as measured by the DFAR was used. In addition, based on independent optical precipitation detector measurements, precipitation had to occur for at least 60% of every 30-min period (18 min). More detailed descriptions of the quality control methods developed and employed in WMO-SPICE are detailed elsewhere (K2017a; Reverdin, 2016).

Following K2017a, a minimum precipitation accumulation threshold for 30-min intervals was identified for every gauge or shield under evaluation to help create an unbiased pool of measurements available for analysis. All 30-min precipitation measurements below the minimum threshold determined for each specific gauge/shield configuration were excluded from the analysis. The minimum precipitation thresholds were calculated by multiplying the minimum DFAR precipitation of 0.25 mm by the median catch efficiency of the gauge under test (for 30-min intervals), using only solid precipitation measurements (mean $T_{air}$ < -2 °C) with high winds (5 m s$^{-1}$ < $U_{10m}$ < 9 m s$^{-1}$). Also following K2017a, a maximum catch efficiency threshold was calculated as 1.0 plus three times the standard deviation of the catch efficiency of the gauge under test, and all measurements exceeding the relevant maximum catch efficiency threshold were excluded from the analysis.

## 2.5 Transfer Function Models

Equation 1 was fit to the quality-controlled 30-minute weighing gauge measurements, following K2017a:

$$CE = e^{-a(U)\left(1 - \tan^{-1}(b(T_{air})) + c\right)} \tag{1}$$

where $CE$ is the catch efficiency, $U$ is the mean wind speed, $T_{air}$ is the mean air temperature, and $a$, $b$, and $c$ are coefficients fit to the data. Equation 2 was also fit to the precipitation measurements, following K2017a:

$$CE = (a)e^{-b(U)} + c \tag{2}$$

where *a*, *b*, and *c* are coefficients fit to the data. Equation 1 was fit as a function of wind speed and air temperature, while Equation 2 was fit separately to solid and mixed precipitation measurements as a function of wind speed only. In the latter case, precipitation type was determined using air temperature, with solid precipitation defined as $T_{air} < -2$ °C, and mixed defined as $2$ °C $\geq T_{air} \geq -2$ °C. These specific temperature thresholds were selected to estimate precipitation type based on past evaluations of precipitation type and air temperature (K2012b, Wolff et al., 2015). For some of the gauges examined here, Eq. 2 unrealistically over-predicted catch efficiency at low wind speeds when insufficiently constrained by the available measurements, and in these cases a more constrained function was used to describe realistic corrections for gauges with fewer low wind speed measurements:

$$CE = (a)e^{-b(U)} + (1 - a), \qquad ` \qquad (3)$$

where *a*, and *b* are coefficients fit to the data. Following K2017a, transfer functions were developed for both gauge height and 10 m height wind speeds.

## 2.6 Maximum wind speed threshold

For the application of transfer functions, a maximum wind speed threshold ($U_{thresh}$) above which the transfer function should not be applied was determined based on a visual assessment of the Eq. 1 transfer function fit to the available measurements. This was done by viewing the catch efficiency function of wind speed and air temperature superimposed on the measurement data, and identifying the wind speed above which all temperature ranges below 2 °C were not generally well represented by the available measurements. The same threshold was applied to the Eq. 2 and Eq. 3 transfer functions. In practice, when the wind speed is above the maximum wind speed threshold, the wind speed should be forced down to the maximum wind speed threshold to adjust the precipitation. A diagram describing the effects of the maximum wind speed threshold on an example transfer function is shown in Figure 3, using the unshielded Eq. 2 type transfer function developed in K2017a.

## 2.7 Testing of transfer functions

When measurements from more than one site were available for a specific gauge or shield, all of the available measurements were merged to create a common transfer function, and the transfer function was then tested on data from each site independently to determine the magnitude of site biases and the appropriateness of the transfer function for each individual site. For gauges that were only tested at one site, a 10-fold cross validation was relied upon to maintain some independence between the measurements used to produce and test the transfer functions. The 10-fold cross validation was performed in 10 separate iterations, each using 90% of the measurements to determine the transfer function and the remaining 10% to test the transfer function. The resulting error statistics were based on the average of all ten iterations.

Errors in the adjusted measurements were estimated by applying the appropriate transfer function and comparing the results to the corresponding DFAR measurements. The errors were then used to calculate the root mean square error (RMSE), the mean bias, the correlation coefficient (*r*), and the percentage of 30 min events with errors less than 0.1 mm ($PE_{0.1\ mm}$). These

statistics were estimated for the Eq. 1 transfer functions using all of the available precipitation measurements. Following K2017a, the Eq. 2 and Eq. 3 transfer function error statistics were estimated by separating the datasets using the mean air temperature into liquid ($T_{air}$ > 2 ˚C), mixed (2 °C ≥ $T_{air}$ ≥ -2 °C.), and solid ($T_{air}$ < -2 ˚C) precipitation, correcting the mixed and solid precipitation using the appropriate transfer functions, combining these results with the uncorrected liquid

precipitation measurements, and comparing the results to the corresponding DFAR measurements.

A t-test with a significance level of 5% was used to evaluate the significance of differences among results adjusted using different transfer functions. When a given precipitation gauge/shield configuration was tested at more than one site, the adjusted measurements from all of the available sites were pooled together before determining the significance of adjusted

measurement differences. The same test was also used to compare unadjusted measurements to adjusted measurements.

## 2.8 Evaluation of independent transfer functions

In addition to developing new adjustments fthe manufacturer-provided weighing gauges tested in WMO-SPICE, the WMO-SPICE weighing gauge measurements were also used to evaluate other independently derived transfer functions that were available. These include the WMO-SPICE single-Alter shielded and unshielded transfer functions derived using eight

different testbeds (K2017a), and transfer functions determined from pre-SPICE measurements recorded at the Marshall testbed within larger windshields (K2017b).

Single-Alter and unshielded transfer functions developed previously for WMO-SPICE host-provided weighing gauges (either Geonor T-200B3 or OTT Pluvio[2]; K2017a) were tested on measurements from all of the manufacturer-provided

unshielded and single-Alter shielded gauges. The hypothesis behind this testing is that the response of a gauge to wind speed and air temperature is more sensitive to wind-shielding (or lack thereof) than to the specific gauge type. Although the transfer functions from K2017a were not developed for these specific gauges, they include measurements from eight different sites. Therefore, they may be more robust and universally applicable than transfer functions developed from measurements at the limited number of sites where a specific manufacturer-provided gauge was tested. A robust transfer

function should arguably be developed from measurements representing a wide variety of precipitation types and wind speeds, as any transfer function is only valid for the range of conditions represented by the measurements used in its development. In addition, as shown in K2017a, site biases do exist, indicating that the use of data from several sites for the creation of a transfer function is preferable to the use of data from just one or two sites. Because of this, it is possible that a generic transfer function developed using data from several sites may be more universally applicable to a manufacturer-

provided weighing gauge than the gauge-specific transfer function. For the double-Alter, Belfort double-Alter, and the SDFIR shielded gauges, which were not tested as broadly within WMO-SPICE as the single-Alter shielded and unshielded gauges, the WMO-SPICE measurements recorded at Marshall and CARE were used to evaluate transfer functions created in K2017b using measurements recorded at the Marshall testbed before WMO-SPICE began.

**2.9 Transfer function uncertainty and wind speed**

Errors in the transfer functions were evaluated further as a function of wind speed by computing RMSE values from the available catch efficiencies after binning by wind speed (1 m s$^{-1}$ bins). For each wind speed bin, the RMSE values of the adjustments were calculated from differences between the transfer function catch efficiencies and the measured catch efficiencies. In addition, RMSE values of the adjusted catch efficiencies were estimated by applying the appropriate transfer function to the measurements and calculating the resultant error in the catch efficiency. This evaluation was limited to solid precipitation ($T_{air} < -2$ ˚C) measurements for ease of presentation, and the Eq. 1 type transfer functions were used for all of the gauge configurations examined.

**3 Results and Discussion**

**3.1 Transfer function types**

Using the methods described above, custom transfer functions were created for all manufacturer-provided weighing gauges and all shield configurations tested in WMO-SPICE (Table 2). For all of the gauges and shields tested, the Eq. 1 transfer function was fit to the calculated catch efficiency values using the measured wind speed and air temperature. In addition, either the Eq. 2 or the Eq. 3 type adjustment was fit to the mixed (2 °C $\geq T_{air} \geq$ -2 °C) and solid ($T_{air} < -2$ ˚C) precipitation measurements as a function of wind speed. These transfer functions were created for both the gauge height wind speed and the 10 m height wind speed. Statistics describing the transfer function errors were calculated based on the differences between the adjusted precipitation measurements and the DFAR precipitation measurements. In addition, transfer functions available from other studies were tested on these precipitation measurements, and errors in the uncorrected measurements were also described.

Based on the results of t-tests used to compare different types of transfer functions, no significant differences were found among measurements adjusted using the Eq. 1, Eq. 2, or Eq. 3 transfer functions. In addition, there were no significant differences between measurements adjusted using the 10 m height wind speeds or the gauge height wind speeds. For the sake of simplicity, and to focus on more significant results, a comparison of results for all transfer functions, for each gauge and shield tested, is not included here. Examples of such comparisons are available in K2017a, and like the present study, these examples demonstrate that the different types of transfer functions fitted to the same data performed similarly. Newly developed and previously existing transfer functions are, therefore, compared using Eq. 1 transfer functions with gauge height winds, with these results found to be representative of all of the different types of transfer functions tested.

### 3.2 Unshielded gauges

Unshielded Sutron, MRW500, and TRwS 405 weighing gauges were tested at several different WMO-SPICE testbeds. Separate transfer functions were developed for each gauge type. In addition, an independent pre-existing 'universal' transfer function (K2017a) was tested on all of the unshielded gauge measurements. Statistics describing errors in the adjusted and unadjusted measurements are shown in Fig. 4. Improvements in the adjusted bias (Fig. 4b) and $PE_{0.1\ mm}$ (Fig. 4d) values were more notable than improvements in the RMSE (Fig. 4a) or correlation coefficients (Fig. 4c). In addition, the 'universal' transfer function performed as well as the transfer functions fitted specifically to measurements from each gauge type. This indicates that all of the unshielded gauges tested, including those from K2017a, will accumulate similar amounts of precipitation when exposed to the same environmental conditions. It also indicates that small differences in the shapes of these gauges do not affect the relationship between their catch efficiency and wind speed. Using t-tests, the measurements adjusted with the 'universal' equation were compared to measurements adjusted with each custom equation. For the unshielded Sutron, MRW500, and TRwS gauges, the t-tests indicated that there were no significant differences between measurements adjusted using gauge-specific transfer functions and the unshielded K2017a transfer function.

### 3.3 Single-shielded gauges

Transfer functions for the single-Alter shielded Sutron and 1500 mm Geonor gauge measurements were developed and tested, in addition to transfer functions for the MRW500 gauge within the small, manufacturer-provided, single-shield (Fig. 5). These results are discussed in more detail in the sections below.

### 3.3.1 Single-Alter shielded gauges

At Marshall, the single-Alter shielded Sutron gauge measurements were adjusted equally as well using both the custom and the 'universal' transfer function from K2017a (Fig. 5 a-d), with a t-test confirming that measurements adjusted using both methods did not differ significantly from each other. This indicates that undercatch for the single-Alter shielded Sutron TPG responds to wind and air temperature in the same manner as the single-Alter shielded gauges used in K2017a.

The 1500 mm Geonor was tested at four different sites, and significant differences were found between the transfer functions fit to these measurements and the K2017a 'universal' transfer function. Figure 5 a-d supports this finding, showing differences between error statistics for measurements adjusted using the Eq. 1 and Univ. Eq. 1 transfer functions. These differences were most apparent in the bias and RMSE values estimated at Weissfluhjoch. For reasons that are not well understood, the catch efficiency of the single-Alter shielded 1500 mm Geonor gauges at Weissfluhjoch and Caribou Creek did not decrease with wind speed as much as at the other sites. The 1500 mm Geonor at Caribou Creek was installed near low trees, which may have sheltered the gauge from the wind from some directions; however, a lack of wind direction measurements available in the WMO-SPICE event dataset from this site prohibits evaluation of this hypothesis. The

Weissfluhjoch site was previously demonstrated to be less sensitive to wind than other sites (K2017a), for reasons that were also difficult to understand or confirm. Using the universal transfer function, the 1500 mm Geonor measurements from Weissfluhjoch were over-corrected (Fig. 5b), whereas using a custom 1500 mm Geonor Eq. 1 adjustment, the resultant bias was much smaller. The significant differences between the 'universal' transfer function and the transfer functions fit to the 1500 mm Geonor measurements can be attributed to differences in the measurements used to create the universal transfer functions and the 1500 mm Geonor transfer functions. Thirty-five percent of the available 1500 mm Geonor measurements were recorded at Weissfluhjoch, where the catch efficiency for all the gauges did not decrease with wind speed to the same degree as most of the other sites. When the number of Weissfluhjoch measurements contributing to the 1500 mm transfer functions was artificially reduced, the resultant 1500 mm transfer function was similar to the universal single-Alter adjustment. At the Marshall and Bratt's Lake sites, where the catch efficiency decreased with wind speed as expected, the universal transfer function performed better than the gauge-specific transfer function.

In general, the different error statistics generated from the 1500 mm Geonor measurements indicate that this gauge was subject to more noise than the host-provided gauges used to develop the universal single-Alter transfer functions in K2017a. For example, at Marshall, the 1500 mm single-Alter Geonor RMSE values were about 0.25 mm and the $PE_{0.1\,mm}$ values were about 60%, while for the 600 mm Geonor at this same site, the RMSE values were about 0.15 mm, and the $PE_{0.1\,mm}$ values were about 70% (K2017a). The increased capacity of the 1500 mm Geonor gauge appears to be associated with a decrease in sensitivity, which may make it more difficult to accurately measure snowfall events and distinguish between gauge noise and light precipitation.

### 3.3.2 Single-shielded MRW500

The MRW500 weighing gauge was provided with a custom single-shield, which was smaller than a standard single-Alter shield and was constructed out of metal slats mounted at a fixed angle (Fig. 2c). Because no previously derived transfer function was available for this specific shield, the 'universal' single-Alter shielded adjustment was tested on this configuration. The resultant RMSE values for the custom and 'universal' transfer functions were similar to each other (Fig. 5e), and the $PE_{0.1}$ values were improved by the use of the 'universal' SA transfer function (Fig 5h). However, the negative bias resulting from the application of the universal single-Alter adjustment indicates that this gauge was generally under-corrected, particularly at Bratt's Lake (Fig. 5f).

Exponential wind speed transfer functions were also developed separately for solid and mixed precipitation. After preliminary analysis, Eq. 3 was developed and used for the shielded MRW500 gauge, because Eq. 2 predicted unreasonably high catch efficiencies as the wind speed approached 0 m s$^{-1}$. This result is probably more closely linked with the scarcity of low wind speed events and random errors in the low wind speed catch efficiencies from these two sites, rather than with the specific gauge configuration. In addition, although an exponential fit was used for these data because it was more realistic at

high wind speeds (where a linear fit would predict negative catch efficiencies), for the data available, the shielded MRW500 catch efficiency responded quite linearly to wind speed. Unfortunately, there were insufficient high-wind data available from the Bratt's Lake and Marshall sites to evaluate this shield at higher winds, where the catch efficiency would presumably reach an asymptote at a minimum value greater than zero. Transfer function coefficients for the MRW500 are available in the Supplement (Table S1).

### 3.4 Double-Alter, Belfort double-Alter, and SDFIR shields

Measurements recorded by gauges within larger shields – the double-Alter, Belfort double-Alter, and SDFIR shields – were subject to smaller uncorrected biases and smaller errors after adjustment. For example, the RMSE values of the adjusted measurements shown in Fig. 6 were smaller than those for the single-Alter and unshielded gauges tested (Fig. 4 and 5), and the $PE_{0.1\,mm}$ values were larger. More specific results are discussed for individual shields below.

### 3.4.1 Double-Alter shield

The double-Alter shield was tested at both the CARE and Marshall testbeds; the CARE site had an OTT Pluvio$^2$ gauge in a double-Alter shield, and the Marshall site had a 600 mm Geonor T-200B3 gauge in a double-Alter shield. The results presented in Figure 6 a-d indicate that the pre-SPICE double-Alter transfer function (K2017b) performed as well as the custom WMO-SPICE transfer function (Fig. 6 a-d). One thousand, three hundred and ninety-two measurements were available from the pre-SPICE Marshall measurements, while only 723 measurements were available from the WMO-SPICE measurements (Table 2). However, this new WMO-SPICE correction is arguably more defensibly applicable to all double-Alter measurements than the pre-SPICE transfer function, because it was developed using measurements from two sites.

### 3.4.2 Belfort double-Alter shield

The Belfort double-Alter shield was tested at both the CARE and Marshall testbeds; the CARE site had an OTT Pluvio$^2$ gauge in a Belfort double-Alter shield, and the Marshall site had a 600 mm Geonor T-200B3 gauge in a Belfort double-Alter shield. The Belfort double-Alter shield was more effective at reducing undercatch than the standard double-Alter shield. This is demonstrated by the generally small RMSE improvements of the corrected measurements over the uncorrected measurements (Fig. 6e), and also by the near-zero uncorrected biases for the gauges at both Marshall and CARE (Fig. 6f). These measurements, recorded at two separate sites, confirm the efficacy of the Belfort double-Alter shield documented by K2017b using measurements from a single site. Nine hundred and nineteen 30-min measurements from gauges in Belfort double-Alter shielded gauges were included in the present WMO-SPICE transfer function development (Table 2), and 1204 30-min measurements were available for the Pre-SPICE Marshall transfer function development (K2017b). Although the two datasets resulted in similar transfer functions, we recommend using the transfer functions determined from the WMO-SPICE measurements, because they include measurements from two sites, and are therefore expected to be more broadly applicable.

### 3.4.2 SDFIR shield

Tested only at the Marshall testbed, the SDFIR was the largest wind shield evaluated in our study, and the uncorrected and corrected SDFIR measurements were associated with the lowest RMSE and bias values (Fig. 6i and 6j), and the highest correlation coefficient and $PE_{0.1\ mm}$ values (Fig. 6k and 6l) among the shields tested. The catch efficiencies determined using the custom WMO-SPICE adjustment functions were similar to those derived from the K2017b SDFIR adjustment, providing independent validation of the pre-SPICE transfer function in K2017b (Fig. 6).

In general, the necessity of transfer function adjustments for SDFIR-shielded measurements is disputable, as the results in Figure 6 demonstrate that the corrected measurements were only marginally better than the uncorrected measurements; only very small improvements were observed in the mean bias values (Fig. 6j) and the $PE_{0.1\ mm}$ values (Fig. 6l). However, errors in the uncorrected SDFIR measurements were determined to be significantly different than zero. These SDFIR-shielded results are also interesting, because they provide a good indication of the magnitude of errors when comparing well-shielded gauges. As such, these measurements are a good representation of the current limits in accuracy for precipitation measurements recorded using two different well-shielded gauges at the same site. The inferences that can be drawn from such well-shielded measurements are further emphasized below in the comparison of the different shields and adjustments.

### 3.5 Synthesis

### 3.5.1 Recommended transfer functions

As shown in Table 3, we recommend using the appropriate K017a transfer functions for all of the unshielded and single-Alter shielded WMO-SPICE gauges. Although the 'universal' single-Alter transfer function performed poorly on measurements from the 1500 mm Geonor gauges at Caribou Creek and Weissfluhjoch, we still recommend using it on measurements from single-Alter shielded 1500 mm Geonor gauges. This is because there is no obvious physical explanation for a higher catch efficiency for the Geonor 1500 mm gauge relative to the 600 mm or 1000 mm Geonor gauges (the collecting area and inlet shape are the same for each configuration). In addition, the relatively poor performance of the 'universal' single-Alter function in this case may be due to the specific population of 1500 mm Geonor measurements available within this intercomparison. As also indicated in Table 3, the new transfer function coefficients provided in the Supplement (Table S1) should be used to correct measurements from MRW500 gauges with the manufacturer-provided single-shield, rather than the K2017a 'universal' single Alter correction that was tested on this gauge/shield combination.

The double-Alter shield transfer function coefficients in Table S2 in the Supplement are recommended for use with measurements from weighing gauges in double-Alter shields. However, due to the limited warm season, liquid precipitation measurements included in the SPICE datasets, the use of the Eq. 1 transfer function coefficients presented here is not recommended when $T_{air}$ is > 5 ˚C, as they produce unrealistically high warm-temperature precipitation catch efficiencies. If

an Eq. 1 type function applicable to warm-season measurements is needed, we recommend using the pre-SPICE function, which performed similarly to the WMO-SPICE functions (Fig. 6 a-d). Like the double-Alter WMO-SPICE transfer function, the Eq. 1 Belfort double-Alter shield transfer function development did not include many liquid precipitation events, but in this case, the resultant transfer functions were more realistic at warm temperatures, and can therefore be recommended for use in all seasons. The associated transfer function coefficients are provided in the Supplement (Table S3).

For the SDFIR, the pre-SPICE and the custom WMO-SPICE transfer functions performed quite similarly (Fig. 6 i-l). The K2017b SDFIR transfer function was developed using five years of measurement data (1508 30-min precipitation events), whereas only two winter seasons (410 30-min events) were available for the development of the WMO-SPICE transfer function. For warm temperature and high wind speed conditions, the lack of rain events in the WMO-SPICE dataset (76 in Table 2) resulted in unrealistically large SDFIR-shielded catch efficiencies predicted by the Eq. 1 type transfer function. The K2017b Eq. 1 type catch efficiencies were more realistic for all temperature/wind speed regimes. For this reason, and because the pre-SPICE transfer function was developed using measurements from the same gauge and shield, at the same site, over a much longer period, we recommend using the Eq. 1 type transfer function from K2017b for measurements from SDFIR-shielded weighing gauges. Because K2017b did not include Eq. 2 coefficients, the Eq. 2 transfer function coefficients determined from the WMO-SPICE measurements are included in Table S4 in the Supplement. The Eq. 2 transfer functions, which are only for solid and mixed precipitation, were unaffected by the lack of warm-temperature precipitation in the WMO-SPICE measurements.

### 3.5.2 Comparison of shield types

Examples of recommended Eq. 1 type adjustments for solid precipitation are included in Fig. 7a, with the transfer functions plotted against the gauge height wind speed with $T_{air}$ = -5 ˚C. The $T_{air}$ value of -5 ˚C was selected because it was fairly representative of the solid precipitation events included in this analysis, which had a median $T_{air}$ of -5.2 ˚C. The unshielded and single Alter 'universal' multi-site Eq. 1 transfer functions from K2017a are also included, as these were generally recommended over the custom gauge-specific unshielded or single-Alter transfer functions developed here. Figure 7a demonstrates the relative magnitudes of the adjustments for different wind shields, with the more effective shields (SDFIR, Belfort double-Alter) resulting in much higher catch efficiencies than less effectively shielded gauges (single-Alter, MRW500 shield) or unshielded gauges.

The uncertainty in each transfer function was also estimated for different wind speeds. For different wind speed bins, transfer function errors were calculated from differences between the measured catch efficiencies and the adjustment (or transfer function) fit to the catch efficiencies. The resultant RMSE values were found to be relatively insensitive to wind speed (Fig. 7b). This is significant, because catch efficiency was presumably less affected by the interaction of snow crystals and wind at low wind speed. This indicates that variability in snowflake habit, which affects hydrometeor drag and fall velocity, may not

be the primary source of uncertainty in the relationship between catch efficiency and wind speed. Other causes of uncertainty, such as random variability in the precipitation gauge measurements and the natural spatial variability of precipitation, may in fact be more important.

In addition, errors in the adjusted catch efficiencies were calculated by applying the appropriate adjustments to the measurements, and calculating RMSE values for the resultant catch efficiencies. After adjustment, the catch efficiency should be equal to approximately 1.0, so the RMSE of the adjusted catch efficiency was quantified using the difference between the adjusted catch efficiency and 1.0 (Fig. 7c). The relationship between the magnitudes of the adjustments and the uncertainties in the adjusted catch efficiencies is apparent from comparison of Figs. 7a and 7c. Measurements that required
larger adjustments experienced larger errors in the adjusted catch efficiencies. This is due, at least in part, to basic arithmetic; for example, a precipitation measurement associated with a predicted catch efficiency of 50% would be doubled by adjustment, and any errors in the measured catch efficiency would likewise be doubled by the adjustment. At a given wind speed, the errors in the adjusted catch efficiencies (Fig. 7c) are approximately equal to the errors in the catch efficiency (Fig. 7b) divided by the adjustment (Fig. 7a). Errors in the measured catch efficiency (shown in Fig. 7b) were enhanced by the
appropriate adjustments (shown in Fig. 7a).

This indicates that despite the necessity and utility of transfer functions, effective wind shielding is recommended and beneficial for the measurement of solid precipitation in windy conditions. Many sites and networks use either unshielded or single-Alter shielded precipitation gauges due to the cost of purchasing, transporting, installing, and maintaining larger
shields. For example, the DFAR used at the Weissfluhjoch site was built in eight sections and flown in piece-by-piece using a helicopter. This level of expense is necessary at an intercomparison site, but may not be feasible on a larger scale within observational networks. Many areas where snowfall is monitored are remote, and cannot be accessed via road. Sufficient space must also be available to install a large wind shield. In addition, wooden shields require maintenance, and will eventually become weathered and require replacement. Such limitations affect meteorological, climate, and hydrological
networks, and must be taken into consideration before selecting a wind shield. For this reason, the performance of the Belfort double-Alter wind shield is notable, as it is much smaller than the SDFIR and DFAR. A stainless steel wind shield like the Belfort double-Alter can also be designed to be relatively low maintenance, as compared to a wooden shield. The efficacy of the Belfort double-Alter shield relative to the standard double-Alter shield also indicates that further improvements in wind shield design are possible by altering the geometry and porosity of the wind shield.

**4 Conclusions**

New transfer functions were developed using precipitation measurements from both host- and manufacturer- provided WMO-SPICE weighing gauges, and were tested alongside existing transfer functions. The resultant errors in corrected

precipitation measurements were presented, and recommendations for the correction of different types of weighing gauges and shield configurations were made. These transfer functions were demonstrated to reduce the mean bias of weighing gauge measurements relative to the DFAR, and the remaining uncertainty in the corrected measurements was described using different statistics.

For the unshielded and single-Alter shielded weighing gauges provided by different manufactures for testing in WMO-SPICE, the 'universal' multi-site transfer function developed in K2017a typically worked as well as the gauge-specific transfer functions developed in this study. Therefore, the more universal unshielded and single-Alter multi-site transfer functions from K2017a are recommended for adjusting measurements from all the unshielded and single-Alter shielded

weighing gauges tested.

The low-porosity double-Alter shield manufactured by Belfort performed well relative to the DFAR, with an average uncorrected bias of only – 0.04 mm, or – 5.4%. The Belfort double-Alter shielded gauges performed better than weighing gauges in traditional single- and double-Alter shields, suggesting that it is a viable, high-efficacy option for networks or sites

that do not have the resources to build, site, and maintain a large wooden shield like the SDFIR or DFIR.

Precipitation measurements from weighing gauges in higher-efficacy shields, such as the SDFIR and the Belfort double-Alter, showed not only much smaller uncorrected biases relative to the corresponding reference configurations, but also smaller adjusted RMSE and higher adjusted $PE_{0.1\ mm}$. Measurements from these gauge/shield configurations required less

adjustment than the unshielded gauges tested, and the resultant errors estimated by comparing the adjusted measurements to the DFAR measurements were also much smaller. The errors that remained after adjusting the unshielded and single-Alter shielded measurements were much larger than the errors experienced by the more effectively shielded gauges. Upon closer inspection and bin-averaging by wind speed, the magnitude of uncorrectable errors in the adjusted measurements increased with the size of the required adjustment; at higher wind speeds, where less-effectively shielded measurements required

doubling or even tripling, the uncertainty in the measurements was also doubled or tripled, accordingly. This suggests that there is a limit to the amount of uncertainty that can be removed by such adjustments, and the transfer functions presented here may already be approaching this limit. These results also suggest that although adjusted unshielded and single-Alter shielded gauge measurements can be used to measure the total amount of precipitation without a large bias, the best way to reduce the uncertainty of the measurement is to shield the gauge more effectively using a shield such as the DFIR, SDFIR, or

Belfort double-Alter.

## 5. Future Work

The transfer functions developed and tested in this study are intended to adjust solid and mixed precipitation measurements recorded by weighing gauges. For other types of winter precipitation sensors, such as disdrometers, present weather sensors, and heated tipping-bucket gauges, further research is needed to develop similar adjustment methods. Transfer functions are also needed for measurements at larger spatial scales; for example, adjustments are needed for gridded global precipitation product generation by national meteorological and climatic networks, and for the validation of global hydrologic, weather, and climate models. With unshielded precipitation gauge measurement errors potentially exceeding 100% of the measured precipitation in windy and cold conditions, the results of such research may be significant, especially in Arctic and mountainous regions.

For the application of these transfer functions, high quality and representative wind speed and air temperature measurements are necessary. Wind speed measurements that accurately characterize the area around the precipitation gauge must be obtained carefully, because small-scale changes in the wind field can be caused by vegetation, buildings, and even the infrastructure of the monitoring site itself (such as solar panels, towers, etc.). The application of transfer functions using gridded measurements will also be affected by errors in wind speed and air temperature, and unresolved changes in surface roughness, local topography, and siting. The effects of such errors would propagate through the transfer function to the resultant catch efficiency, so the adjusted precipitation must also be examined. Such work can be performed using high-quality reference precipitation measurements for validation, which may additionally reveal climate biases in the transfer functions. Since a limited number of WMO-SPICE sites with a DFAR configuration were available to develop the transfer functions, not all climate types were well represented. For example; no maritime or Arctic sites were included in WMO-SPICE. The work described here constitutes an important step forward in the development and evaluation of transfer functions for precipitation measurements, and it also presents opportunities for further improvements.

### Acknowledgements

The authors thank Hagop Mouradian from Environment and Climate Change Canada for contributing the mapped site locations (Fig. 1). We thank the manufacturers that provided many of the sensors used to produce these results. We also thank the World Meteorological Organization for supporting this intercomparison. In addition, Howard Diamond, from the Atmospheric Turbulence and Diffusion Division of NOAA's Air Resources Laboratory, helped obtain the US and European climate data used to estimate the percentage of solid precipitation shown in Figure 1.

## Disclaimers

Many of the results presented in this work were obtained as part of WMO-SPICE, conducted on behalf of the World Meteorological Organization (WMO) Commission for Instruments and Methods of Observation (CIMO). The analysis and views described herein are those of the authors at this time, and do not necessarily represent the official outcome of WMO-

SPICE. Mention of commercial companies or products is solely for the purposes of information and assessment within the scope of the present work, and does not constitute a commercial endorsement of any instrument or instrument manufacturer by the authors or the WMO.

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

| Gauge | Measurement | Heater | Output | Capacity | Recorded rate |
|---|---|---|---|---|---|
| MPS Systém TRwS 405 | Load cell | 15 W | Digital | 750 mm | 1 min (Ma, Hauk) |
| Meteoservis MRW500 | Load cell | 350 W | Digital | 900 mm | 1 min (Ma, BrLa) |
| Sutron TPG | Load cell | 200 W | Digital | 914 mm | 6 s (Ma) |
| 1500 mm Geonor | Vibrating wire | 60 W (Weis), N/A* (BrLa, CaCr, Ma) | Analog | 1500 mm | 6 s (Ma, Weis), 1 min (BrLa, CaCr) |
| 600 mm Geonor | Vibrating wire | 60 W | Analog | 600 mm | 6 s (Ma) |
| OTT Pluvio[2] | Load cell | 50 W | Digital | 1500 mm | 6 s (CARE) |

**Table 1. Gauge type, measurement type, heater power, output type, gauge capacity, and the recorded rate of the different types of gauges included in this evaluation. The recorded rate was determined by the logging system at a given site, and was not intrinsic to the gauges. Testbeds included in the study include the Centre for Atmospheric Research Experiments (Canada, abbr. CARE), Caribou Creek (Canada, abbr. CaCr), Bratt's Lake (Canada, abbr. BrLa), Marshall (United States, Ma), Haukeliseter (Norway, abbr. Hauk), and Weissfluhjoch (Switzerland, abbr. Weis). *The 1500 mm Geonor gauges at Bratt's Lake, Caribou Creek, and Marshall were tested without heaters.**

| Shield | Gauge | Testbed | Mean $U_{gh}$ | Max $U_{gh}$ | $N_{liquid}$ | $N_{mixd}$ | $N_{solid}$ |
|---|---|---|---|---|---|---|---|
| Unshielded | Sutron TPG | Ma | 2.8 m s$^{-1}$ | 10.2 m s$^{-1}$ | 77 | 166 | 208 |
| Unshielded | MRW500 | Ma, BrLa | 3.2 m s$^{-1}$ | 10.2 m s$^{-1}$ | 121 | 214 | 230 |
| Unshielded | TRwS 405 | Ma, Hauk | 4.0 m s$^{-1}$ | 17.0 m s$^{-1}$ | 128 | 250 | 250 |
| MRW500 shield | MRW500 | Ma, BrLa | 3.2 m s$^{-1}$ | 10.2 m s$^{-1}$ | 121 | 214 | 230 |
| Single Alter | Sutron TPG | Ma | 2.8 m s$^{-1}$ | 10.2 m s$^{-1}$ | 78 | 172 | 201 |
| Single Alter | 1500 mm Geonor | Ma, BrLa, Weis, CaCr | 3.3 m s$^{-1}$ | 11.6 m s$^{-1}$ | 172 | 374 | 996 |
| Double Alter | Geonor/Pluvio[2] | Ma, CARE | 3.0 m s$^{-1}$ | 10.2 m s$^{-1}$ | 147 | 173 | 403 |
| Belfort double Alter | Geonor/Pluvio[2] | Ma, CARE | 3.0 m s$^{-1}$ | 10.2 m s$^{-1}$ | 206 | 244 | 496 |
| SDFIR | Geonor | Ma | 2.9 m s$^{-1}$ | 10.2 m s$^{-1}$ | 76 | 173 | 161 |

**Table 2. Shield type, gauge type, testbed, mean and maximum (Max) of the measured gauge height wind speed ($U_{gh}$), and the number of liquid ($N_{liquid}$), mixed ($N_{mixd}$), and solid ($N_{solid}$) 30-min precipitation measurements included in this study. The measurements for all gauges were recorded during the two winter seasons (Oct 1 – April 30) of 2013-2015. Wind speed statistics only describe periods of precipitation.**

| Shield | Gauge | Recommendation |
|---|---|---|
| Unshielded | Sutron TPG | Tables 2 – 3 in K2017a |
| Unshielded | MRW500 | Tables 2 – 3 in K2017a |
| Unshielded | TRwS 405 | Tables 2 - 3 in K2017a |
| Unshielded | 1500 mm Geonor | Tables 2 - 3 in K2017a |
| MRW500 shield | MRW500 | Table S1 in Supplement |
| Single Alter | Sutron TPG | Tables 2 - 3 in K2017a |
| Double Alter | Geonor/Pluvio[2] | Table S2 in Supplement |
| Belfort double Alter | Geonor/Pluvio[2] | Table S3 in Supplement |
| SDFIR | Geonor | Table 2 in K2017b; Table S4 in Supplement |

**Table 3. Recommended transfer functions for different weighing precipitation gauges and shields.**

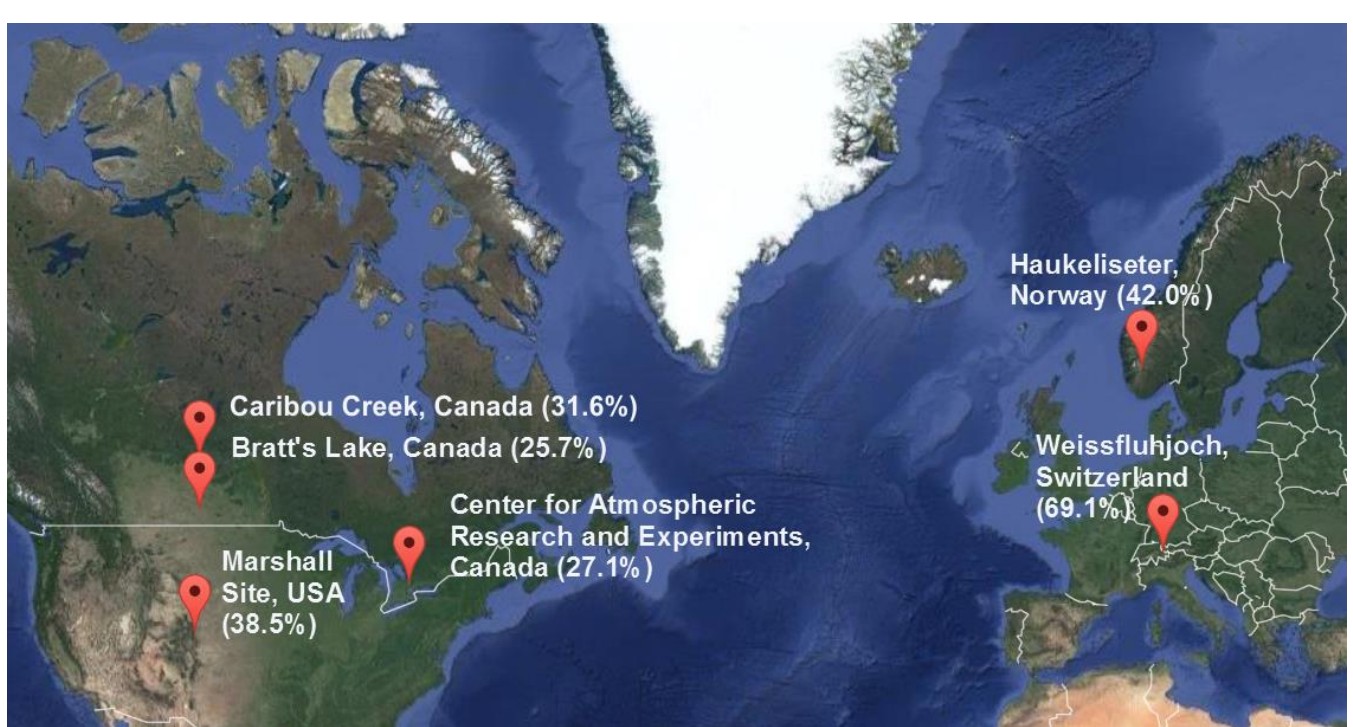

**Figure 1. Map of WMO-SPICE test sites with weighing gauges considered in this study. In addition, the percent of total precipitation that occurred as solid precipitation is shown in brackets. For the Caribou Creek, Bratt's Lake, Center for Atmospheric Research and Experiments (CARE), and Marshall sites the closest sites with 30 year Climate Normals (1981-2010) were used; at Haukeliseter and at Weissfluhjoch local observations were used from 1994 - 2017 and 1981 - 2010, respectively.**

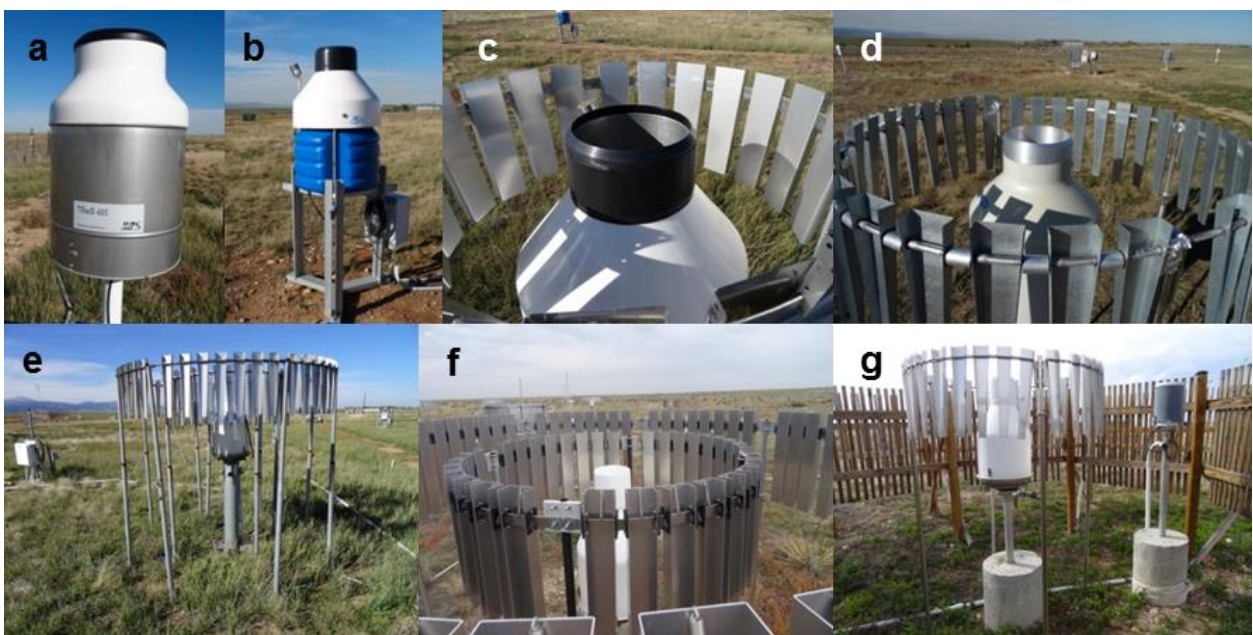

**Figure 2. Photos of the (a) TRwS 405, (b) unshielded MRW500, (c) shielded MRW500, (d) single-Alter shielded Sutron, (e) double-Alter shielded Geonor, (f) Belfort double-Alter shielded Geonor, and (g) SDFIR-shielded Geonor at the Marshall, CO, US testbed.**

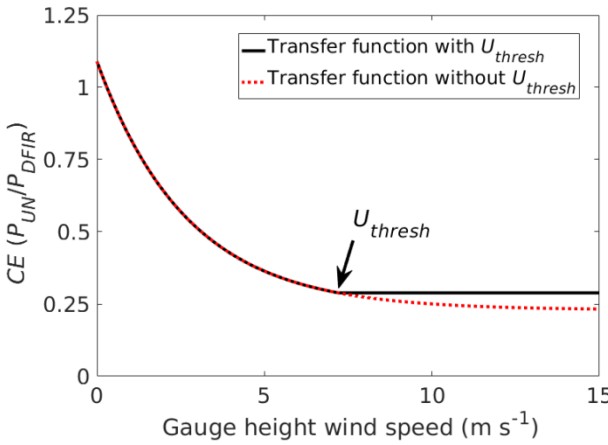

5    **Figure 3. Example application of the maximum wind speed threshold ($U_{thresh}$), using an Eq. 2 type transfer function describing unshielded (UN) solid precipitation catch efficiency (*CE*) from K2017a. At wind speeds exceeding $U_{thresh}$ (7.2 m s$^{-1}$ in this case) the catch efficiency is fixed at the value determined at $U_{thresh}$.**

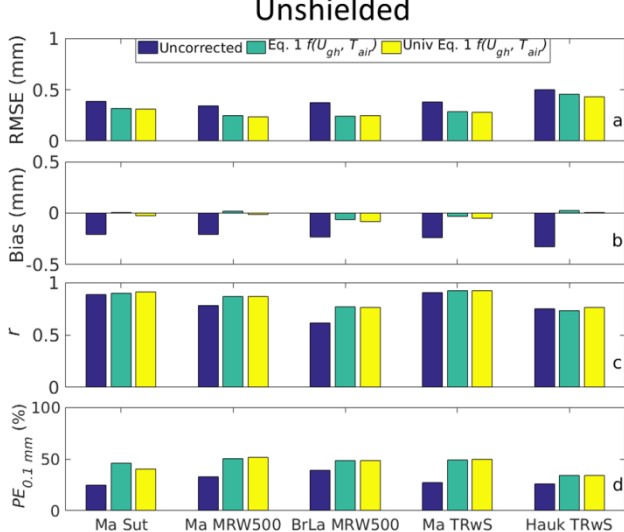

**Figure 4. (a) Root mean square error (RMSE), (b) bias, (c) correlation coefficient (*r*), and (d) the percentage of events with errors less than 0.1 mm (*PE$_{0.1\ mm}$*) calculated from the difference between the 30 min precipitation measurements from the corrected, unshielded gauges and the DFAR. The Sutron from Marshall (Ma Sut), the MRW500 from Marshall (Ma MRW500) and Bratt's Lake (BrLa MRW500), and the TRwS 405 from Marshall (Ma TRWS) and Haukeliseter (Hauk TRWS) are included. Uncorrected measurements are also shown (Uncorrected, dark blue). Measurements adjusted using an example transfer function (Eq. 1) fit to precipitation, air temperature (*T$_{air}$*) and gauge height wind speed (*U$_{gh}$*) are shown in green. Statistics describing these measurements adjusted using the 'universal' unshielded transfer function (Univ. Eq. 1, yellow), which were based on independent Eq. 1 coefficients derived in K2017a, are also shown.**

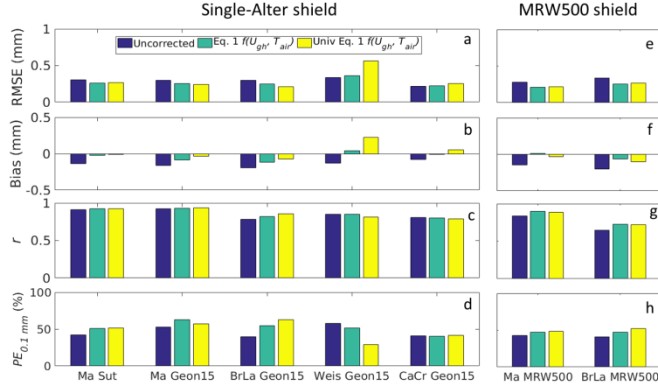

**Figure 5. Error statistics calculated for the single-Alter shielded Marshall Sutron gauge (Ma Sut) and single-Alter shielded 1500 mm Geonor gauges (Geon15) at the Marshall (Ma), Bratt's Lake (BrLa), Weissfluhjoch (Weis), and Caribou Creek (CaCr) testbeds (a – d). Error statistics from the shielded MRW500 gauges from Marshall (Ma MRW500) and Bratt's Lake (BrLa MRW500) are also shown (e – h). The different statistics and correction types are described in the Fig. 4 caption. The results of the 'universal' single-Alter transfer function (Univ. Eq. 1, yellow), which were based on independent Eq. 1 coefficients derived in K2017a, are shown for all of the single-Alter shielded gauges and also for the gauges within the MRW500 shield.**

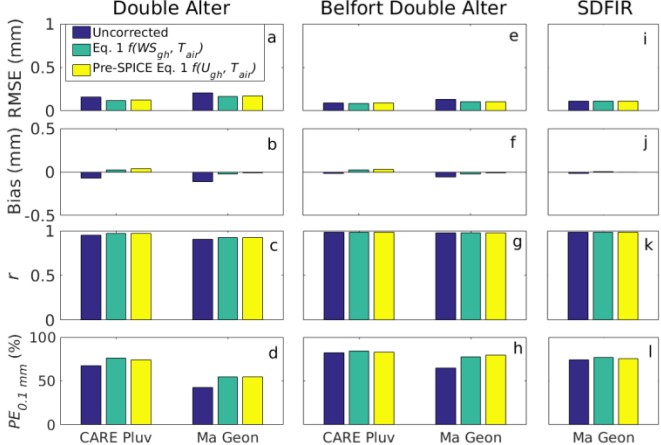

**Figure 6.** Error statistics calculated from the double-Alter shielded Pluvio[2] gauge at CARE (Care Pluv, a - d), the double-Alter shielded Geonor at Marshall (Ma Geon, a - d), the Belfort double-Alter shielded Pluvio[2] at CARE (CARE Pluv, e – f), the Belfort double-Alter shielded Geonor at Marshall (Ma Geon, e – h), and the SDFIR shielded Geonor at Marshall (Ma Geon, i – l). The different adjustment types are described in the Fig. 4 caption. Statistics describing these measurements adjusted using the appropriate pre-SPICE transfer functions from K2017b are also shown (Pre-SPICE. Eq. 1, yellow).

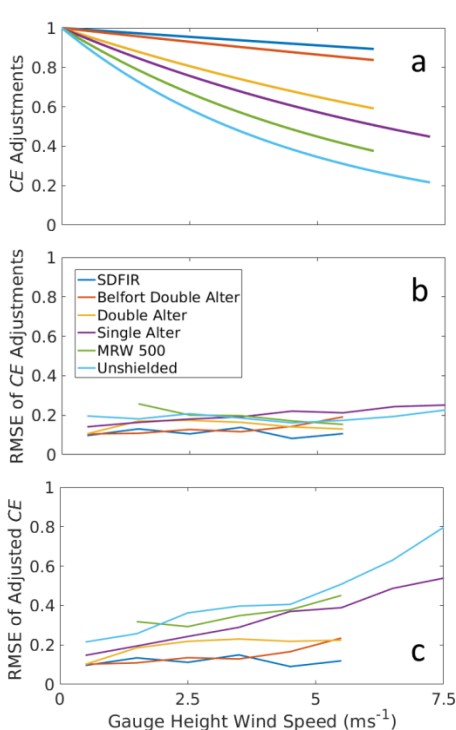

**Figure 7.** (a) Comparison of recommended Eq. 1 catch efficiency (*CE*) adjustments at $T_{air}$ = -5 °C. (b) RMSE of the *CE* adjustments, estimated by bin-averaging for every 1 m s$^{-1}$ of wind speed. (c) RMSE of the adjusted *CE* bin-averaged by wind speed, estimated by adjusting the measurements and calculating RMSE values from the resultant catch efficiencies. The SDFIR, Belfort double- Alter, standard double-Alter and MRW500 transfer function coefficients were determined from the results presented here. The single-Alter shielded and unshielded results are from K2017a. Bin-averaged RMSE values are only shown (b and c) when more than 30 values were available within a given 1 m s$^{-1}$ wind speed range.