# Peer review of "Testing and Development of Transfer Functions for Weighing Precipitation Gauges in WMO-SPICE"

_Hydrology and Earth System Sciences, 2017_

## Referee Comment (RC1) · Anonymous Referee #1 · 6 Jul 2017

The manuscript "Testing and Development of transfer functions for weighing precipitation gauges in WMO-SPICE" deals with the interesting and relevant topic of how to correct undercatch especially of snowfall for different sensors. This paper is basically an extension of the Kochendorfer et al. 2017 (HESS) that again is based on the (Kochendorfer et al. 2016) study. Here the apply the previously developed correction functions to different sensors and locations and found that the previously developed, more general correction function of Kochendorfer et al. 2017 (HESS) is always recommended to use. In general the paper is mostly very written and I found no crucial error or mistake. My major concern is more related to presentation and the structure of the manuscript. Both section "Methods" and "Results (& Discussion)" are written very detailed, technically and lengthy, giving each sensor, its location, and treatment of the data its place. Figures 4 - 11 display in detail the (mostly non-existing) differences between correction functions. Partly, I found it hard to follow the overall structure/story of the manuscript. In my opinion, this structure would have been justified if the authors would have found several different best-performing equations that need to be presented alongside each sensor and in comparison with the general correction function. But, given that in most cases the general (somehow site and sensor unspecific) correction function is as well performing and recommended by the authors, a more summarizing structure would have been much more feasible and would have cause less redundancies. In its current version, the manuscript unfortunately reads quite lengthy, especially given the final outcome. Many of the interpretations named in the abstract and conclusion are a bit out of blue and should be much more elaborated. Hence, instead of the detailed description, I would recommend to show some summarizing, and comparing analyses/graphics on sensor performance.

Based on this basic concern, and the following further smaller concerns, I recommend to provide either a revised, restructured version that strongly condense the specific sensor part and put more focus on the comparison analysis, or to show the transferability of the Kochendorfer et al. 2017a Equation 3 to other sensors in a HESS-technical note. I am sorry to be that harsh, but I really do think that the level of detail and length of the manuscript does not match your findings.

Further general concerns:

1. Several time differences are named significant or non-significant, but I somehow missed the section describing how a significance test was performed.

2. Given that both the biases of the corrected values and the differences between the corrected functions are mostly rather small (Figure 4- 11, RMSE < 0.5 mm, biases « 0.5 mm, differences far less), I wonder how and if at one can and should interpret these differences, given the measurement accuracy of each sensor.

3. The paper needs to be more independent from Kochendorfer et al. 2017 (HESS). At least the essential equation 3 should also be given in this paper.

Further specific comments:

page 2, line 19-22: I do not understand this sentence. Please check language and try to avoid too long sentences.

page 3ff: I would recomment to use Table 1 much more often to guide the reader through the different configurations.

page 3, A paragraph that outlines the design of this study would help the reader to follow your following descriptions

page 3, line 9: typo: "a either a"

page 3, line 31: CARE, acronym not yet introduced

page 4, line 4, "have" missing in and a lower porosity

page 4, line 31: 6 s and 1 min data correspond to with sensor / site. Maybe update Table 1

page 4, line 32: "realistic" and "operational" limits, please define

page 5, Maybe I missed it, but where do you define that DFAR is the reference, and please discuss briefly its quality and its deviation from the "truth"

page 5, line 13ff: How were temperature thresholds defined. If possible refer to citations

page 5, line 18: Please provide these equation also here to be more independent from the Kochendorfer et al. (2007a) paper

page 7, line 13ff: I do not understand this interpretation. If significant site biases exists, doesn't that mean that you have to develop site-specific empirical correction functions?

page 10, line 31: "1500mm" Geonor ? cp. with page 11, line 1: please be consistent, see also page 11, line 17

page 11, line 6: typo: "attributed"

page 11, line 20: does the noise of the Geonor 1500 mm stem from the fact that they have a different sensitivity than the 500 mm ?

page 11, line 20 ff: Are these interpretation valid, given the some deviation and the measurement accuracy?

page 11, line 26, CARE and Marshall *test sites* !

page 17, Table 1: please add time period of measurements considered

---

## Referee Comment (RC2) · Anonymous Referee #2 · 15 Jul 2017

The paper presents results from tests on the adequacy of (mostly already existing) transfer functions for precipitation gauges. It directly builds on the paper by Kochendorfer et al. 2017, recently published in HESS.

The topic discussed by the Authors is of scientific relevance and timely, and its scope is within the objectives of HESS. The manuscript presents novel findings that may be useful to inform the selection of proper instrumentation for measuring (solid) precipitation. Results and conclusions are clearly outlined; however, I believe the overall presentation should be substantially restructured to better convey the manuscript's findings. Specifically, I think that, in its current form, the manuscript lacks important pieces of

information and an overall picture that would enhance its comprehension.

In the following, I report a few suggestions for improvement.

General comments:

1. A short introduction should be provided on the reasons why new precipitation gauges are needed, what are the criticalities in measuring precipitation, what we expect the improvements from using alternative measurement systems would be. I understand that many of these aspects were already outlined by the Authors in Kochendorfer et al. 2017; herein, the Authors should focus on measurement equipment alternative to traditional systems. Alternative precipitation gauging systems are blossoming in the hydrological community, and their promise/limitations may be reported to better support the scope of the paper and expand the bibliography.

2. The role of wind in the underestimation of precipitation should be better highlighted through key citations.

3. Why were these specific gauging systems selected? I think that the description of gauges can be improved by providing further details on how they work, what their features contribute to, and what we should be expecting in terms of performance and limitations. I also suggest that Figure 2 is improved and key features are highlighted for each of the gauges.

4. The Discussion and Conclusions should clearly state what research findings are and recommend best practice for measuring solid precipitation. I suggest the Authors include a Table in the "Synthesis" section where each gauge is coupled with the recommended transfer functions and comments are provided on eventual limitations.

Specific comments:

1. Abstract: I think the Abstract should be simplified (it is not necessary to list the names of all gauges) and the paper objectives and results clearly outlined.

[Figure]

2. Introduction: I believe including a synthesis Table on previous experiments would help the reader to frame the work within previous studies. I also recommend the Authors expand the last paragraph by (i) justifying the selection of specific gauges; (ii) clearly stating hypotheses; (iii) and identifying key objectives.

3. Methods: Many parameters/terms were not properly defined. I believe the Authors should devote a paragraph to re-state what catch efficiency is, what are key variables influencing the response of the gauges, and to report previously developed transfer functions. Please also clarify the data structure (sentence on Page 4 line 31 is out of the blue).

4. Results and Discussion: Since many of the tested gauges had a similar behavior, I do not think separate sections and Figures 4 to 12 are necessary. I suggest the Authors consolidate results in a Table. I would also move the transfer function coefficients in the Supplementary Material.

5. I think the presentation quality of the paper is sufficient; however, the number of references could be extended. I also suggest the Authors double check the English for minor typos. I herein list some of them:

- Page 3 line 9: "consisted of a either a"

- Page 10 line 6: "This result are"

- Page 11 line 6: "measurements are attributed"

- Page 13 line 17: "3-dimensional" (please clarify what this means)

- Page 15 line 4: This sentence is unclear, please elaborate.

I also suggest the Authors pay special attention in defining all acronyms.
* * *

---

## Author Comment (AC1) · 4 Sep 2017

**General comments from Referee #1:**

The manuscript "Testing and Development of transfer functions for weighing precipitation gauges in WMO-SPICE" deals with the interesting and relevant topic of how to correct undercatch especially of snowfall for different sensors. This paper is basically an extension of the Kochendorfer et al. 2017 (HESS) that again is based on the (Kochendorfer et al. 2016) study. Here the apply the previously developed correction functions to different sensors and locations and found that the previously developed, more general correction function of Kochendorfer et al. 2017 (HESS) is always recommended to use. In general the paper is mostly very written and I found no crucial error or mistake. My major concern is more related to presentation and the structure of the manuscript. Both section "Methods" and "Results (& Discussion)" are written very detailed, technically and lengthy, giving each sensor, its location, and treatment of the data its place. Figures 4 - 11 display in detail the (mostly non-existing) differences between correction functions. Partly, I found it hard to follow the overall structure/story of the manuscript. In my opinion, this structure would have been justified if the authors would have found several different best-performing equations that need to be presented alongside each sensor and in comparison with the general correction function. But, given that in most cases the general (somehow site and sensor unspecific) correction function is as well performing and recommended by the authors, a more summarizing structure would have been much more feasible and would have cause less redundancies. In its current version, the manuscript unfortunately reads quite lengthy, especially given the final outcome. Many of the interpretations named in the abstract and conclusion are a bit out of blue and should be much more elaborated. Hence, instead of the detailed description, I would recommend to show some summarizing, and comparing analyses/graphics on sensor performance.

Based on this basic concern, and the following further smaller concerns, I recommend to provide either a revised, restructured version that strongly condense the specific sensor part and put more focus on the comparison analysis, or to show the transferability of the Kochendorfer et al. 2017a Equation 3 to other sensors in a HESS-technical note. I am sorry to be that harsh, but I really do think that the level of detail and length of the manuscript does not match your findings.

**Authors' response to general comments:**

We think this is a fair assessment of the manuscript in some respects. We chose to separate the different types of WMO-SPICE weighing gauge measurements into different manuscripts before discovering that the 'universal' transfer functions (Kochendorfer et al., 2017a) performed well on all of the manufacturer-provided single-Alter shielded and unshielded gauges. The success of the 'universal' transfer functions on these other weighing gauges was a surprise. We had originally assumed that all of the different types of WMO-SPICE weighing gauges would require their own unique transfer function, and it was based in part on this assumption that we decided to separate the WMO-SPICE weighing gauge transfer functions into two separate manuscripts. In addition, the length and amount of detail included in the Kochendorfer et al. (2017a) manuscript would have been unwieldy if all of the weighing gauges from WMO-SPICE were added to it.

We agree that the mode of presenting the results can be simplified and improved. For example, many of the figure styles were developed for Kochendorfer et al. (2017a), and were chosen to comprehensively compare different types of transfer functions using different wind speeds. In order to achieve a more condensed view, we will change the figure style by merging gauges into the same plot and excluding some of the extraneous transfer functions. As the reviewer pointed out, the majority of the transfer functions evaluated in this manuscript were determined to be obsolete because they were no better than an appropriate 'universal' transfer function or a pre-SPICE transfer function from Kochendorfer et al. (2017b). In addition, for a given gauge the differences between any of the appropriate transfer functions evaluated were quite small. Because of this, they do not all merit such a thorough evaluation, the format

of the results can indeed be consolidated significantly in response to the reviewer's suggestion. This will help to improve the presentation of the manuscript and to focus it on the more significant results.

We maintain, however, that the results are significant enough to merit publication as a new manuscript. The manuscript provides independent validation of previously-existing transfer functions, and it also demonstrates that transfer functions derived using one type of weighing precipitation gauge can be used on another type of similar gauge. Many readers (including gauge and site developers, precipitation observers and data users) will find this significant, surprising, and also very useful. The manuscript, based on a well-executed and carefully-designed experiment, includes the evaluation of nine different types of gauges and wind shields.

Reviewer #1 should also note that in addition to the different single-Alter and unshielded gauges that were evaluated, gauges in several other types of windshields were also assessed. Some measurements recorded within these other winds shields were used to validate the results of Kochendorfer et al. (2017b), and others were shown to merit new adjustments. For example, a new set of adjustments for the shielded MRW500 precipitation gauge were derived and recommended. Arguably, such an evaluation on its own could merit publication as a full manuscript, as this is the only transfer function available for this gauge and shield so far, but due to the wealth of new measurements produced by the WMO-SPICE project, it was included with the evaluation of transfer functions for all of the other WMO-SPICE weighing precipitation gauges and wind shields.

The team of WMO-SPICE investigators who authored this manuscript has an obligation to precipitation gauge manufacturers and their customers to recommend transfer functions for each weighing gauge and wind shield tested in the WMO-SPICE project. In addition to the forthcoming WMO report, a journal article like this is the best way to disseminate the results of an international project like WMO-SPICE.

**Further general concerns from Referee #1:**

1. Several time differences are named significant or non-significant, but I somehow missed the section describing how a significance test was performed.

**Authors' response:** In response to this valid comment, T-tests have been performed to more objectively evaluate the significance of differences between the errors associated with different transfer functions and also the unadjusted measurements. The results of these T-tests will be documented in the revised manuscript, and will be used to replace the subjective determinations of significance included in the original manuscript.

2. Given that both the biases of the corrected values and the differences between the corrected functions are mostly rather small (Figure 4- 11, RMSE < 0.5 mm, biases « 0.5 mm, differences far less), I wonder how and if at one can and should interpret these differences, given the measurement accuracy of each sensor.

**Authors' response:** For the most part, the biases in the corrected measurements and the differences between the different corrections are indeed negligible. The restructured manuscript will state this more explicitly.

3. The paper needs to be more independent from Kochendorfer et al. 2017 (HESS). At least the essential equation 3 should also be given in this paper.

**Authors' response:** This is a good suggestion. The often-cited Eq. 3 will be included independently in the paper, and other changes will be made to help the manuscript stand on its own.

Further specific comments:

page 2, line 19-22: I do not understand this sentence. Please check language and try to avoid too long sentences.

**Authors' response:** The run-on sentence will be divided up for clarification.

page 3ff: I would recomment to use Table 1 much more often to guide the reader through the different configurations.

**Authors' response:** There are indeed many different gauges and configurations for the reader to familiarize himself or herself with, and the suggestion is a good one. We will try to refer to Table 1 throughout this section.

page 3, A paragraph that outlines the design of this study would help the reader to follow your following descriptions

**Authors' response:** We will add an introductory paragraph describing how the intercomparison was designed, with a common automated reference gauge at all sites, along with other shared types of precipitation and meteorological measurements. The overarching goal of the intercomparison was to include as many different countries, climates, and gauges as possible while still maintaining some basic standards.

page 3, line 9: typo: "a either a"

**Authors' response:** Thank you! This will be corrected.

page 3, line 31: CARE, acronym not yet introduced

**Authors' response:** CARE is "Centre for Atmospheric Research Experiments", and will be defined in the manuscript.

page 4, line 4, "have" missing in and a lower porosity

**Authors' response:** This will be rewritten.

page 4, line 31: 6 s and 1 min data correspond to with sensor / site. Maybe update Table 1

**Authors' response:** We will describe which data are available with which sensors.

page 4, line 32: "realistic" and "operational" limits, please define

**Authors' response:** We will clarify this. The term "realistic" was meant to signify "possible", and was used to remove periods of calibrations/validations and other impossibly high-rate precipitation and negative measurements from the record. The term, "operational" was meant to signify "within the operational specifications of the sensor", such as a weighing gauge depth that was beyond the upper limit of the gauge specifications.

page 5, Maybe I missed it, but where do you define that DFAR is the reference, and please discuss briefly its quality and its deviation from the "truth"

**Authors' response:** The DFAR was described and defined as the reference at the beginning of the Methods section. The "truth" is indeed difficult to determine for snowfall because pit gauge measurements, which are used as a reference for rain, are subject to blowing snow and capping, and are therefore not appropriate for snowfall. The DFIR is the reference for manual snowfall observations (Goodison et al., 1998.; Yang, 2014). In the beginning of the Methods section, we will add available references to manual DFIR and "bush" gauges, in addition to later comparisons with automated bush gauges. We will also discuss in greater detail how and why the DFAR was defined as the reference for WMO-SPICE.

page 5, line 13ff: How were temperature thresholds defined. If possible refer to citations

**Authors' response:** The temperature thresholds were defined based on histograms of automated precipitation type and air temperature measurements (Kochendorfer et al., 2017b; Wolff et al., 2015). An explanation and citations will be added to the manuscript.

page 5, line 18: Please provide these equation also here to be more independent from the Kochendorfer et al. (2007a) paper

**Authors' response:** This is a good suggestion.

page 7, line 13ff: I do not understand this interpretation. If significant site biases exists, doesn't that mean that you have to develop site-specific empirical correction functions?

**Authors' response:** This is indeed true, but we do not have a good way to determine such site-specific empirical corrections. These adjustments are designed for use at sites that do not have a good reference, and at such sites there is no good way to determine what the site bias is. For example, in Kochendorfer et al. (2017a) the three high-altitude sites had the largest biases, but at two of these high-altitude sites the biases were negative, and at the other high-altitude site the bias was positive.

page 10, line 31: "1500mm" Geonor ? cp. with page 11, line 1: please be consistent, see also page 11, line 17

**Authors' response:** That is correct. We will correct these inconsistencies.

page 11, line 6: typo: "attributed"

**Authors' response:** Thank you. The spelling error will be corrected.

page 11, line 20: does the noise of the Geonor 1500 mm stem from the fact that they have a different sensitivity than the 500 mm ?

**Authors' response:** Yes that is probably the case. We will suggest this in the manuscript.

page 11, line 20 ff: Are these interpretation valid, given the some deviation and the measurement accuracy?

**Authors' response:** We may disagree on the definition of 'noise', but it is certainly the case that it is more difficult to measure light snowfall with a 1500 mm Geonor than a 600 mm Geonor. In general, it can be difficult to differentiate signal noise from precipitation with these weighing gauges, and this problem is augmented with the 1500 mm Geonor. This is an important issue for the measurement of

snowfall (and light rain over shorter time periods, such as 5 min) because most snowfall is associated with low precipitation rates, particularly in polar regions.

page 11, line 26, CARE and Marshall *test sites* !

**Authors' response:** Thank you. This will be corrected.

page 17, Table 1: please add time period of measurements considered

**Authors' response:** The time period will be documented in the Table 1 caption rather than the actual table. All of the measurements spanned the same time period, so it isn't necessary to describe the time period in the table for each individual site.

**References**

Kochendorfer, J., Nitu, R., Wolff, M., Mekis, E., Rasmussen, R., Baker, B., Earle, M. E., Reverdin, A., Wong, K., Smith, C. D., Yang, D., Roulet, Y. A., Buisan, S., Laine, T., Lee, G., Aceituno, J. L. C., Alastrué, J., Isaksen, K., Meyers, T., Brækkan, R., Landolt, S., Jachcik, A., and Poikonen, A.: Analysis of single-Alter-shielded and unshielded measurements of mixed and solid precipitation from WMO-SPICE, Hydrol. Earth Syst. Sci., 21, 3525-3542, 2017a.

Kochendorfer, J., Rasmussen, R., Wolff, M., Baker, B., Hall, M. E., Meyers, T., Landolt, S., Jachcik, A., Isaksen, K., Brækkan, R., and Leeper, R.: The quantification and correction of wind-induced precipitation measurement errors, Hydrol. Earth Syst. Sci., 21, 1973-1989, 2017b.

WMO Solid Precipitation Measurement Intercomparison (1998, WMO/TD-No. 872, IOM No. 67, by B.E. Goodison, P.Y.T. Louie and D. Yang)

Wolff, M. A., Isaksen, K., Petersen-Overleir, A., Odemark, K., Reitan, T., and Braekkan, R.: Derivation of a new continuous adjustment function for correcting wind-induced loss of solid precipitation: results of a Norwegian field study, Hydrology and Earth System Sciences, 19, 951-967, 2015.

Yang, D, 2014: Double Fence Intercomparison Reference (DFIR) vs. Bush Gauge for "true" snowfall measurement, Journal of Hydrology,Vol. 509, p. 94-100

---

## Author Comment (AC2) · 4 Sep 2017

**Overview from Reviewer #2**

The paper presents results from tests on the adequacy of (mostly already existing) transfer functions for precipitation gauges. It directly builds on the paper by Kochendorfer et al. 2017, recently published in HESS.

The topic discussed by the Authors is of scientific relevance and timely, and its scope is within the objectives of HESS. The manuscript presents novel findings that may be useful to inform the selection of proper instrumentation for measuring (solid) precipitation. Results and conclusions are clearly outlined; however, I believe the overall presentation should be substantially restructured to better convey the manuscript's findings. Specifically, I think that, in its current form, the manuscript lacks important pieces of information and an overall picture that would enhance its comprehension.

**Authors' response:** Thank you. We agree with this evaluation and will restructure the manuscript based on the reviewer's suggestions.

In the following, I report a few suggestions for improvement.

**General comments:**

1. A short introduction should be provided on the reasons why new precipitation gauges are needed, what are the criticalities in measuring precipitation, what we expect the improvements from using alternative measurement systems would be. I understand that many of these aspects were already outlined by the Authors in Kochendorfer et al. 2017; herein, the Authors should focus on measurement equipment alternative to traditional systems. Alternative precipitation gauging systems are blossoming in the hydrological community, and their promise/limitations may be reported to better support the scope of the paper and expand the bibliography.

**Authors' response:** The Introduction will be augmented with a description of why new precipitation gauges are needed, and areas where the authors see the potential for improvements. By 'alternative' precipitation measurement systems, we assume that Reviewer #2 means non-catchment types of gauges, which include the hotplate and optical devices such as the present weather sensors, present weather detectors, disdrometers, and optical rain gauges. Some of these types of gauges were included in WMO-SPICE, and the results will be detailed in the forthcoming project report. A detailed discussion of these gauges is beyond the scope of the present manuscript, which focusses on weighing gauges ('traditional' systems).

2. The role of wind in the underestimation of precipitation should be better highlighted through key citations.

**Authors' response:** This is a good suggestion; we will add more citations describing the effects of wind on gauge catch efficiency.

3. Why were these specific gauging systems selected? I think that the description of gauges can be improved by providing further details on how they work, what their features contribute to, and

what we should be expecting in terms of performance and limitations. I also suggest that Figure 2 is improved and key features are highlighted for each of the gauges.

**Authors' response:** All of the weighing precipitation gauges tested in WMO-SPICE have been included in the manuscript. Many of the gauges were provided by the manufacturers that chose to participate in WMO-SPICE, and others were provided (and selected) by site hosts for their own national and scientific interests. Such an explanation will be added to the manuscript. Some description of the individual gauges was provided in the appropriate Results sections, but these will be expanded and moved to the Methods section. This may also help with the restructuring of the manuscript.
Figure 2 was included to provide readers with visual examples of field installations, and to help familiarize readers with the different types of gauges and windshields discussed in the manuscript. Such images can also be used to visually assess the effects of wind on the different types of gauges and shields. These examples provide valuable context for the interpretation of results and discussion regarding wind effects on different gauge/shield combinations. The technical features of the gauges will be described in a new table, added to the revised manuscript. However visual depictions of the specific transducers for each gauge type, and of other gauge elements such as heaters and buckets, are not critical to the interpretation of results, and will not be included.

4. The Discussion and Conclusions should clearly state what research findings are and recommend best practice for measuring solid precipitation. I suggest the Authors include a Table in the "Synthesis" section where each gauge is coupled with the recommended transfer functions and comments are provided on eventual limitations.

**Authors' response:** This is an excellent suggestion. We will modify and restructure the manuscript, which will have a larger Synthesis Section, and a smaller Section describing the results of the individual gauges. A Table will be added clearly documenting the recommended transfer functions for each gauge and shield.

**Specific comments:**

1. Abstract: I think the Abstract should be simplified (it is not necessary to list the names of all gauges) and the paper objectives and results clearly outlined.

**Authors' response:** We will remove the list of gauges and further simplify the Abstract by focusing on the general objectives and results.

2. Introduction: I believe including a synthesis Table on previous experiments would help the reader to frame the work within previous studies. I also recommend the Authors expand the last paragraph by (i) justifying the selection of specific gauges; (ii) clearly stating hypotheses; (iii) and identifying key objectives.

**Authors' response:** These are good suggestions. The Introduction will be expanded to describe in greater detail how the present work relies upon and supports previous studies. Clarification of

the inclusion of all available WMO-SPICE weighing gauges will be included. Hypothesis and key objectives will also be described.

3. Methods: Many parameters/terms were not properly defined. I believe the Authors should devote a paragraph to re-state what catch efficiency is, what are key variables influencing the response of the gauges, and to report previously developed transfer functions. Please also clarify the data structure (sentence on Page 4 line 31 is out of the blue).

**Authors' response:** Thank you. The Introduction will be expanded to more thoroughly introduce the reader to new terminology, precipitation gauge undercatch, and transfer functions. The Methods section will be augmented with definitions of key terms, measurements, and the data structure.

4. Results and Discussion: Since many of the tested gauges had a similar behavior, I do not think separate sections and Figures 4 to 12 are necessary. I suggest the Authors consolidate results in a Table. I would also move the transfer function coefficients in the Supplementary Material.

**Authors' response:** These are good suggestions. We will develop a more succinct way to present the main results, and can move the transfer function coefficients to the Supplementary Material.

5. I think the presentation quality of the paper is sufficient; however, the number of references could be extended. I also suggest the Authors double check the English for minor typos. I herein list some of them:

**Authors' response:** Thank you. We will double check for minor typos and add more references to the Introduction.

- Page 3 line 9: "consisted of a either a"

**Authors' response:** This will be corrected.

- Page 10 line 6: "This result are"

**Authors' response:** This will be corrected.

- Page 11 line 6: "measurements are attributed"

**Authors' response:** This will be corrected.

- Page 13 line 17: "3-dimensional" (please clarify what this means)

**Authors' response:** Thank you. By '3-dimensional' we meant as a function of both wind speed and air temperature. The term will be removed, as it is unnecessary.

- Page 15 line 4: This sentence is unclear, please elaborate.

**Authors' response:** The following sentence clarifies: "At higher wind speeds, where such measurements require doubling or even tripling, the uncertainty in the measurements was also doubled or tripled, accordingly." But the sentence is indeed confusing, and will be rewritten.

I also suggest the Authors pay special attention in defining all acronyms

**Authors' response:** The manuscript will be reviewed carefully to ensure that all acronyms have been defined properly.

---

## Author Response (AR1)

**Response to Referee #1**

**General comments from Referee #1:**

The manuscript "Testing and Development of transfer functions for weighing precipitation gauges in WMO-SPICE" deals
with the interesting and relevant topic of how to correct undercatch especially of snowfall for different sensors. This paper is
basically an extension of the Kochendorfer et al. 2017 (HESS) that again is based on the (Kochendorfer et al. 2016) study.
Here the apply the previously developed correction functions to different sensors and locations and found that the previously
developed, more general correction function of Kochendorfer et al. 2017 (HESS) is always recommended to use. In general
the paper is mostly very written and I found no crucial error or mistake. My major concern is more related to presentation
and the structure of the manuscript. Both section "Methods" and "Results (& Discussion)" are written very detailed,
technically and lengthy, giving each sensor, its location, and treatment of the data its place. Figures 4 - 11 display in detail
the (mostly non-existing) differences between correction functions. Partly, I found it hard to follow the overall
structure/story of the manuscript. In my opinion, this structure would have been justified if the authors would have found
several different best-performing equations that need to be presented alongside each sensor and in comparison with the
general correction function. But, given that in most cases the general (somehow site and sensor unspecific) correction
function is as well performing and recommended by the authors, a more summarizing structure would have been much more
feasible and would have cause less redundancies. In its current version, the manuscript unfortunately reads quite lengthy,
especially given the final outcome. Many of the interpretations named in the abstract and conclusion are a bit out of blue and
should be much more elaborated. Hence, instead of the detailed description, I would recommend to show some summarizing,
and comparing analyses/graphics on sensor performance.

Based on this basic concern, and the following further smaller concerns, I recommend to provide either a revised,
restructured version that strongly condense the specific sensor part and put more focus on the comparison analysis, or to
show the transferability of the Kochendorfer et al. 2017a Equation 3 to other sensors in a HESS-technical note. I am sorry to
be that harsh, but I really do think that the level of detail and length of the manuscript does not match your findings.

**Authors' response to general comments:**

We think this is a fair assessment of the manuscript in some respects. We chose to separate the different types of WMO-
SPICE weighing gauge measurements into different manuscripts before discovering that the 'universal' transfer functions
(Kochendorfer et al., 2017a) performed well on all of the manufacturer-provided single-Alter shielded and unshielded
gauges. The success of the 'universal' transfer functions on these other weighing gauges was a surprise. We had originally
assumed that all of the different types of WMO-SPICE weighing gauges would require their own unique transfer function,
and it was based in part on this assumption that we decided to separate the WMO-SPICE weighing gauge transfer functions
into two separate manuscripts. In addition, the length and amount of detail included in the Kochendorfer et al. (2017a)
manuscript would have been unwieldy if all of the weighing gauges from WMO-SPICE were added to it.

We agree that the mode of presenting the results can be simplified and improved, and have made changes accordingly. For
example, many of the figure styles were developed for Kochendorfer et al. (2017a), and were chosen to comprehensively
compare different types of transfer functions using different wind speeds. In order to achieve a more condensed view, we
have changed the figure style by merging gauges into the same plot and excluding some of the transfer functions. As the
reviewer pointed out, the majority of the transfer functions evaluated in this manuscript were determined to be obsolete
because they were no better than an appropriate 'universal' transfer function or a pre-SPICE transfer function from
Kochendorfer et al. (2017b). In addition, the differences between most of the transfer functions evaluated were quite small.
Because of this, they do not all merit such a thorough evaluation, the format of the results has been consolidated significantly

in response to the reviewer's suggestion. This has helped to improve the presentation of the manuscript and to focus it on the more significant results.

We maintain, however, that the results are significant enough to merit publication as a new manuscript. The manuscript provides independent validation of previously-existing transfer functions, and it also demonstrates that transfer functions derived using one type of weighing precipitation gauge can be used on another type of similar gauge. Many readers (including gauge and site developers, precipitation observers and data users) will find this significant, surprising, and also very useful. The manuscript, based on a well-executed and carefully-designed experiment, includes the evaluation of nine different types of gauges and wind shields.

Reviewer #1 should also note that in addition to the different single-Alter and unshielded gauges that were evaluated, gauges in several other types of windshields were also assessed. Some measurements recorded within these other winds shields were used to validate the results of Kochendorfer et al. (2017b), and others were shown to merit new adjustments. For example, a new set of adjustments for the shielded MRW500 precipitation gauge were derived and recommended. Arguably, such an evaluation on its own could merit publication as a full manuscript, as this is the only transfer function available for this gauge and shield so far, but due to the wealth of new measurements produced by the WMO-SPICE project, it was included with the evaluation of transfer functions for all of the other WMO-SPICE weighing precipitation gauges and wind shields.

The team of WMO-SPICE investigators who authored this manuscript has an obligation to precipitation gauge manufacturers and their customers to recommend transfer functions for each weighing gauge and wind shield tested in the WMO-SPICE project. In addition to the forthcoming WMO report, a journal article like this is the best way to disseminate the results of an international project like WMO-SPICE.

**Further general concerns from Referee #1:**

1. Several time differences are named significant or non-significant, but I somehow missed the section describing how a significance test was performed.

**Authors' response:** In response to this t-tests have been performed to more objectively evaluate the significance of differences between the errors associated with different transfer functions and also the unadjusted measurements. The results of these t-tests have been documented in the revised manuscript to replace the subjective determination of significance included in the original manuscript.

2. Given that both the biases of the corrected values and the differences between the corrected functions are mostly rather small (Figure 4- 11, RMSE < 0.5 mm, biases « 0.5 mm, differences far less), I wonder how and if at one can and should interpret these differences, given the measurement accuracy of each sensor.

**Authors' response:** For the most part, the biases in the corrected measurements and the differences between the different corrections are indeed negligible. The restructured manuscript states this more explicitly.

3. The paper needs to be more independent from Kochendorfer et al. 2017 (HESS). At least the essential equation 3 should also be given in this paper.

**Authors' response:** This is a good suggestion. The often-cited Eq. 3 and Eq. 4 have been included independently in the paper, and other changes have been made to help the manuscript stand on its own.

Further specific comments:

page 2, line 19-22: I do not understand this sentence. Please check language and try to avoid too long sentences.

**Authors' response:** The run-on sentence has been divided up for clarification.

page 3ff: I would recomment to use Table 1 much more often to guide the reader through the different configurations.

**Authors' response:** There are indeed many different gauges and configurations for the reader to familiarize himself or herself with, and the suggestion is a good one. We now refer to the table (which is now Table 2) throughout this section.

page 3, A paragraph that outlines the design of this study would help the reader to follow your following descriptions

**Authors' response:** We have added an introductory paragraph describing how the intercomparison was designed, with a common automated reference gauge at all sites, along with other shared types of precipitation and meteorological measurements. The goal was to include as many different countries, climates, and gauges as possible while still maintaining some basic standards.

page 3, line 9: typo: "a either a"

**Authors' response:** Thank you! This has been corrected.

page 3, line 31: CARE, acronym not yet introduced

**Authors' response:** CARE is "Centre for Atmospheric Research Experiments", has been defined in the manuscript.

page 4, line 4, "have" missing in and a lower porosity

**Authors' response:** This has been rewritten.

page 4, line 31: 6 s and 1 min data correspond to with sensor / site. Maybe update Table 1

**Authors' response:** We have now described which data are available with which sensors in a new Table 1.

page 4, line 32: "realistic" and "operational" limits, please define

**Authors' response:** We have clarified this in the manuscript. The term "realistic" was meant to signify "possible", and was used to remove periods of calibrations/validations and other impossibly high-rate precipitation measurements from the record. The term, "operational" was meant to signify "within the operational specifications of the sensor", such as a weighing gauge depth that was beyond the upper limit of the gauge specifications.

page 5, Maybe I missed it, but where do you define that DFAR is the reference, and please discuss briefly its quality and its deviation from the "truth"

**Authors' response:** The DFAR was described and defined as the reference at the beginning of the Methods section. The "truth" is indeed difficult to determine for snowfall because pit gauge measurements, which are used as a reference for rain, are subject to blowing snow and capping, and are therefore not appropriate for snowfall. The DFIR is the reference for manual snowfall observations (Goodison et al., 1998.; Yang, 2014). In the beginning of the Methods section, we have added available references to manual DFIR and "bush" gauges, in addition to later comparisons with automated bush gauges if references are available. We also have added a more detailed discussion of how and why the DFAR was defined as the reference for WMO-SPICE.

page 5, line 13ff: How were temperature thresholds defined. If possible refer to citations

**Authors' response:** The temperature thresholds were defined based on histograms of automated precipitation type and air temperature measurements (Kochendorfer et al., 2017b; Wolff et al., 2015). An explanation and citations have been added to the manuscript.

page 5, line 18: Please provide these equation also here to be more independent from the Kochendorfer et al. (2007a) paper

**Authors' response:** The equations have been added to the manuscript.

page 7, line 13ff: I do not understand this interpretation. If significant site biases exists, doesn't that mean that you have to develop site-specific empirical correction functions?

**Authors' response:** This is indeed true, but we do not have a good way to determine such site-specific empirical corrections. These adjustments are designed for use at sites that do not have a good reference, and at such sites there is no good way to determine what the site bias is. For example, in Kochendorfer et al. (2017a) the three high-altitude sites had the largest biases, but at two of these high-altitude sites the biases were negative, and at the other high-altitude site the bias was positive.

page 10, line 31: "1500mm" Geonor ? cp. with page 11, line 1: please be consistent, see also page 11, line 17

**Authors' response:** These inconsistencies have been corrected.

page 11, line 6: typo: "attributed"

**Authors' response:** The spelling error has been corrected.

page 11, line 20: does the noise of the Geonor 1500 mm stem from the fact that they have a different sensitivity than the 500 mm ?

**Authors' response:** Yes that is probably the case. We now suggest this in the manuscript.

page 11, line 20 ff: Are these interpretation valid, given the some deviation and the measurement accuracy?

**Authors' response:** We may disagree on the definition of 'noise', but it is certainly the case that it is more difficult to measure light snowfall with a 1500 mm Geonor than a 600 mm Geonor. In general, it can be difficult to differentiate signal noise from precipitation with these weighing gauges, and this problem is augmented with the 1500 mm Geonor. This is an important issue for the measurement of snowfall (and light rain over shorter time periods, such as 5 min) because most snowfall is associated with low precipitation rates, particularly in the Arctic.

page 11, line 26, CARE and Marshall *test sites* !

**Authors' response:** Thank you. This has been corrected.

page 17, Table 1: please add time period of measurements considered

**Authors' response:** The time period has been documented in the Table 2 caption rather than the actual table. All of the measurements spanned the same time period, so it isn't necessary to describe the time period in the table for each individual site.

**Authors' response:** All of the weighing precipitation gauges tested in WMO-SPICE were included in the manuscript. Many of the gauges were provided by the manufacturers that chose to participate in WMO-SPICE, and others were provided (and selected) by site hosts for their own national and scientific interests. Such an explanation has been added to the manuscript. Some description of the individual gauges was provided in the appropriate Results sections, but these have been expanded and moved to the Methods section. This also helped with the restructuring of the manuscript.
Figure 2 was included to provide readers with visual examples of field installations, and to help familiarize readers with the different types of gauges and windshields discussed in the manuscript. Such images can also be used to visually assess the effects of wind on the different types of gauges and shields. These examples provide valuable context for the interpretation of results and discussion regarding wind effects on different gauge/shield combinations. The technical features of the gauges are now described in a new table, added to the revised manuscript. However visual depictions of the specific transducers for each gauge type, and of other gauge elements such as heaters and buckets, are not critical to the interpretation of results, and will not be included.

4. The Discussion and Conclusions should clearly state what research findings are and recommend best practice for measuring solid precipitation. I suggest the Authors include a Table in the "Synthesis" section where each gauge is coupled with the recommended transfer functions and comments are provided on eventual limitations.

**Authors' response:** We have modified and restructured the manuscript accordingly, with a larger Synthesis Section, and a smaller Section describing the results of the individual gauges. A table has also been added referencing the recommended transfer functions available either in the Supplemental Material or earlier manuscripts.

**Specific comments:**

1. Abstract: I think the Abstract should be simplified (it is not necessary to list the names of all gauges) and the paper objectives and results clearly outlined.

**Authors' response:** We have removed the list of gauges and further simplified the Abstract by focusing on the general objectives and results.

2. Introduction: I believe including a synthesis Table on previous experiments would help the reader to frame the work within previous studies. I also recommend the Authors expand the last paragraph by (i) justifying the selection of specific gauges; (ii) clearly stating hypotheses; (iii) and identifying key objectives.

**Authors' response:** The Introduction has been expanded to describe in greater detail how the present work relies upon and supports previous studies. Clarification of the inclusion of all available WMO-SPICE weighing gauges has been included. Hypothesis and key objectives have also been described.

3. Methods: Many parameters/terms were not properly defined. I believe the Authors should devote a paragraph to re-state what catch efficiency is, what are key variables influencing the response of the gauges, and to report previously developed transfer functions. Please also clarify the data structure (sentence on Page 4 line 31 is out of the blue).

**Authors' response:** Thank you. The Introduction has been expanded to more thoroughly introduce the reader to new terminology, precipitation gauge undercatch, and transfer functions. The Methods section has been augmented with definitions of key terms, measurements, and the data structure.

4. Results and Discussion: Since many of the tested gauges had a similar behavior, I do not think separate sections and Figures 4 to 12 are necessary. I suggest the Authors consolidate results in a Table. I would also move the transfer function coefficients in the Supplementary Material.

**Authors' response:** These are good suggestions. We developed a more succinct way to present the main results, and moved the transfer function coefficients to the Supplement.

5. I think the presentation quality of the paper is sufficient; however, the number of references could be extended. I also suggest the Authors double check the English for minor typos. I herein list some of them:

**Authors' response:** The number of references has been increased, and several of the authors have checked the revised manuscript for typos

- Page 3 line 9: "consisted of a either a"has been

**Authors' response:** This has been corrected.

- Page 10 line 6: "This result are"

**Authors' response:** This has been corrected.

- Page 11 line 6: "measurements are attributed"

**Authors' response:** This has been corrected.

- Page 13 line 17: "3-dimensional" (please clarify what this means)

**Authors' response:** By '3-dimensional' we meant as a function of both wind speed and air temperature. The term has been removed, as it is unnecessary.

- Page 15 line 4: This sentence is unclear, please elaborate.

**Authors' response:** The following sentence clarifies: "At higher wind speeds, where such measurements require doubling or even tripling, the uncertainty in the measurements was also doubled or tripled, accordingly." But the sentence is indeed confusing, and has been rewritten.

I also suggest the Authors pay special attention in defining all acronyms

**Authors' response:** The manuscript has been reviewed carefully to ensure that all acronyms have been defined properly.

[revised manuscript text omitted]

---

## Referee Report (RR1)

Review of HESS-2017-228
Testing and Development of Transfer Functions for Weighing Precipitation Gauges
in WMO-SPICE
John Kochendorfer et al.

Dear editor,

I have assessed the latest version of the manuscript and generally agree with
the previous reviewers that the manuscript is well written. Furthermore, I did
not find significant errors or mistakes.
I share the concern with the previous reviewers that the current manuscript is
an extension of two previous papers in which the transfer functions were
determined. Therefore, I suggest to highlight the implications of these findings
for global solid precipitation measurements in order to make the results more
relevant for the HESS audience.

Sincerely,
Obbe Tuinenburg

nb.: If necessary, you can contact me at O.A.Tuinenburg@uu.nl for the scripts to
create the figures.
* * *
Comments:
1. Please define CE before its use in equation 1.

2. P11,L27-28: Would it be possible (data wise) to use the wind speed
variability to test this hypothesis?

3. More in general, how do these functions hold if the wind speeds are very
variable?

4. To highlight the broader context of this research, maybe Figure 1 can be
adapted to show the fraction of solid precipitation in a shading. As for example
the fraction in ERA-interim precipitation that is snow.

5. Again to stress the large scale applicability of this research, maybe a last
figure could be added to the manuscript in which a typical global distribution
of CE values is given for actual temperature and wind data. (See example figure
for ERA-Interim for 2002, where the CE values (weighed for snowfall) are given
for a sample gauge, based on Equation 1.) In that way, the reader knows what the
scale of the error is if these transfer functions have not been applied. In this
case this error is up to 25% over land, which may be quite significant for some
applications.

[Figure]

*ERA-Interim fraction of precipitation that falls as snow (2002)*

[Figure]

*Values for CE, weighed by amount of snow for one gauge, based on ERA-Interim (2002)*

---

## Author Response (AR2)

**Response to Obbe Tuinenburg**

Comments:

1. Please define CE before its use in equation 1.

**From the authors:** An abbreviated *CE* has been added after the first use of 'Catch Efficiency', on P 4, Ln. 30.

2. P11,L27-28: Would it be possible (data wise) to use the wind speed variability to test this hypothesis?

**From the authors:** There is only one wind speed measurement available at this site, so the spatial variability of wind speed at the site cannot be analysed. In addition, the wind speed measurements were low-frequency, so they cannot be used to evaluate how turbulent or laminar the flow was.

3. More in general, how do these functions hold if the wind speeds are very variable?

**From the authors:** This is a good question. The commonly-held view is that the main requirement for the transfer function is that the wind speed be representative of conditions during precipitation. For example, when adjusting a 24 h precipitation measurement, if the mean 24 h wind speed is not representative of the period of time when the precipitation actually occurred, this will of course affect the suitability of the adjustment (Goodison et al., 1998). More to the point, while developing transfer functions based on 1 h precipitation measurements, Wolff et al. (2015) excluded periods with variable wind speeds from their analysis in an effort to reduce the scatter and uncertainty of the resultant catch efficiencies. Although they retained this extra step in their final analysis, it did not have a significant effect on the catch efficiency or the uncertainty of the transfer function (personal communication).

4. To highlight the broader context of this research, maybe Figure 1 can be adapted to show the fraction of solid precipitation in a shading. As for example the fraction in ERA-interim precipitation that is snow.

**From the authors:** The fraction of solid precipitation for each of the sites has been determined using available climate measurements, and these fractions have been added to Figure 1. An example figure showing the percent of solid precipitation is shown below in Figure R1. Such analyses can be refined further for addition to the manuscript if necessary, but as the focus of this manuscript is not on the regional distribution of solid precipitation or the evaluation of gridded precipitation measurements, we would prefer to exclude it.

5. Again to stress the large scale applicability of this research, maybe a last figure could be added to the manuscript in which a typical global distribution of CE values is given for actual temperature and wind data. (See example figure for ERA-Interim for 2002, where the CE values (weighed for snowfall) are given for a sample gauge, based on Equation 1.) In that way, the reader knows what the scale of the error is if these transfer functions have not been applied. In this case this error is up to 25% over land, which may be quite significant for some applications.

**From the authors:** The plots the reviewer showed are a nice end-point to this work, and we agree that such figures help demonstrate the significance of the recommended transfer functions. Unfortunately there was a general agreement among the authors that we cannot support the application of the transfer functions on a global scale, following Tuinenburg's nice demonstration. In our varied disciplines within our varied specialties, our comfort levels with different amounts of uncertainty vary. It may be a matter of aesthetics to some extent, but as measurement specialists we cannot defend the application of the transfer functions to global precipitation datasets. One of the challenges is in obtaining a global wind and temperature dataset of sufficient quality to be relevant across the globe. We maintain that such an analysis has to be done at a local scale taking into account topography, siting, etc. In addition, not all precipitation measurements are recorded with weighing gauges; the extent to which alternative measurements, such as snow depth, tipping bucket, and remotely sensed precipitation measurements contribute to global precipitation records will affect the applicability of the weighing gauge transfer functions developed here. Such analyses are beyond the scope of this manuscript, but a new Section on Future Work has been added to the end of the manuscript. This Section briefly describes how the upscaling of such transfer functions should be performed, and is included in the response below:

**5. Future Work**

The transfer functions developed and tested in this study are intended to adjust solid and mixed precipitation measurements recorded by weighing gauges. For other types of winter precipitation sensors, such as disdrometers, present weather sensors, and heated tipping-bucket gauges, further research is needed to develop similar adjustment methods. Transfer functions are also needed for measurements at larger spatial scales; for example, adjustments are needed for gridded global precipitation product generation by national meteorological and climatic networks, and for the validation of global hydrologic, weather, and climate models. With unshielded precipitation gauge measurement errors potentially exceeding 100% of the measured precipitation in windy and cold conditions, the results of such research may be significant, especially in Arctic and mountainous regions.

For the application of these transfer functions, high quality and representative wind speed and air temperature measurements are necessary. Wind speed measurements that accurately characterize the area around the precipitation gauge must be obtained carefully, because small-scale changes in the wind field can be caused by vegetation, buildings, and even the infrastructure of the monitoring site itself (such as solar panels, towers, etc.). The application of transfer functions using gridded measurements will also be affected by errors in wind speed and air temperature, and unresolved changes in surface roughness, local topography, and siting. The effects of such errors would propagate through the transfer function to the resultant catch efficiency, so the adjusted precipitation must also be examined. Such work can be performed using high-quality reference precipitation measurements for validation, which may additionally reveal climate biases in the transfer functions. Since a limited number of WMO-SPICE sites with a DFAR configuration were available to develop the transfer

functions, not all climate types were well represented. For example; no maritime or Arctic sites were included in WMO-SPICE. The work described here constitutes an important step forward in the development and evaluation of transfer functions for precipitation measurements, and it also presents opportunities for further improvements.

5 **Response to Reviewer #2**

The Authors have addressed all of my comments and the manuscript is now suitable to be accepted in HESS.

**From the authors:** Thank you very much!

Before publication, I recommend double-checking the manuscript for typos. See, for instance, line 19 (...and weighing
10 gauges typically measure...)

**From the authors:** This typo has been corrected, and the entire manuscript has been checked for remaining typos.

**Figures**

[Figure]

**Figure R1. The ratio of solid precipitation to total precipitation, based on the 1981-2010 climate normals (1759 locations across Canada). The values are showin in %.**

[revised manuscript text omitted]